# Structure of the MlaC-MlaD complex reveals molecular basis of periplasmic phospholipid transport

Peter Wotherspoon [1,7], Hannah Johnston [1,7], David J. Hardy[1,7], Rachel Holyfield [1,7], Soi Bui [2,3,7], Giedrė Ratkevičiūtė [1,4,7], Pooja Sridhar [1], Jonathan Colburn [5], Charlotte B. Wilson [1], Adam Colyer [1], Benjamin F. Cooper [1,6], Jack A. Bryant[1], Gareth W. Hughes [1], Phillip J. Stansfeld [5], Julien R. C. Bergeron [2] & Timothy J. Knowles [1] ✉

The Maintenance of Lipid Asymmetry (Mla) pathway is a multicomponent system found in all gram-negative bacteria that contributes to virulence, vesicle blebbing and preservation of the outer membrane barrier function. It acts by removing ectopic lipids from the outer leaflet of the outer membrane and returning them to the inner membrane through three proteinaceous assemblies: the MlaA-OmpC complex, situated within the outer membrane; the periplasmic phospholipid shuttle protein, MlaC; and the inner membrane ABC transporter complex, MlaFEDB, proposed to be the founding member of a structurally distinct ABC superfamily. While the function of each component is well established, how phospholipids are exchanged between components remains unknown. This stands as a major roadblock in our understanding of the function of the pathway, and in particular, the role of ATPase activity of MlaFEDB is not clear. Here, we report the structure of *E. coli* MlaC in complex with the MlaD hexamer in two distinct stoichiometries. Utilising in vivo complementation assays, an in vitro fluorescence-based transport assay, and molecular dynamics simulations, we confirm key residues, identifying the MlaD β6-β7 loop as essential for MlaCD function. We also provide evidence that phospholipids pass between the C-terminal helices of the MlaD hexamer to reach the central pore, providing insight into the trajectory of GPL transfer between MlaC and MlaD.

Gram-negative bacteria are typified by a multi-layer cell envelope architecture composed of three principal components: the outer membrane (OM), the periplasm (containing peptidoglycan cell wall) and the inner membrane (IM). Whilst the IM is primarily composed of glycerophospholipids (GPLs), the OM has an asymmetric arrangement, with lipopolysaccharides (LPS) exclusively located in the outer leaflet and GPLs in the inner leaflet. Dysregulation of OM lipid localisation disrupts the cell envelope's ability to function as an effective

[1]School of Biosciences, University of Birmingham, Birmingham, UK. [2]Randall Centre for Cell & Molecular Biophysics, School of Basic & Medical Biosciences, King's College London, London, UK. [3]Charles River Laboratories, 8-9 The Spire Green Centre, Harlow, UK. [4]Department of Biochemistry, University of Oxford, Oxford, UK. [5]School of Life Sciences and Department of Chemistry, University of Warwick, Coventry, UK. [6]Sir William Dunn School of Pathology, University of Oxford, Oxford, UK. [7]These authors contributed equally: Peter Wotherspoon, Hannah Johnston, David J. Hardy, Rachel Holyfield, Soi Bui, Giedrė Ratkevičiūtė. ✉e-mail: t.j.knowles@bham.ac.uk

permeability barrier against a wide variety of molecules, including antimicrobial agents[1]. Whilst the mechanism of LPS transport is well established[2], GPL trafficking between the IM and OM requires further characterisation[3–5].

The Maintenance of outer membrane Lipid Asymmetry (Mla) pathway has been identified as being involved in safeguarding the asymmetric lipid arrangement of the OM, by removing ectopic GPLs from the outer leaflet of the OM and returning them to the IM[6]. The pathway consists of an OM lipoprotein component (MlaA), found as part of a complex with either OmpC or OmpF[7,8]; a periplasmic GPL chaperone (MlaC) which shuttles GPL across the periplasm[9]; and an IM ATP binding cassette (ABC) transporter complex (MlaFEDB)[10]. MlaFEDB consists of four proteins: MlaE an ABC permease protein; MlaD a membrane anchored MCE (mammalian cell entry) domain containing protein forming a homo-hexameric ring that faces the periplasm; MlaF, an ABC ATPase; and MlaB, a STAS (Sulfate Transporter and Anti-Sigma factor antagonist) domain protein proposed to have a regulatory function[11].

Cryo-Electron Microscopy (cryo-EM) structures of the *E. coli, A. Baumannii*, and *P. aeruginosa* MlaFEDB complexes have been resolved in various nucleotide- and substrate-bound states[12–17], providing insight into the workings of the MlaFEDB complex. In addition, the crystal structure of the *E. coli* MlaC has been reported in isolation as well as bound to GPL molecules[9,18]. However, the molecular details of the interaction between MlaFEDB and MlaC have, until recently, remained elusive. Ercan et al. attempted to map this interaction through cross-linking experiments[19], and a more detailed understanding of the interaction has only recently been proposed[20], with the assistance of AlphaFold2 predictions. Although, due to the simulated nature of this model and the low-resolution of the actual structural data presented there is still a large degree of uncertainty regarding the nature of the MlaFEDB to MlaC interaction.

In this study, we address this knowledge gap. Specifically, we report the structure of *E. coli* MlaC in complex with the MlaD hexamer, in two different stoichiometries (1:6 and 2:6, respectively), stabilised through the binding of cardiolipin. Utilising in vivo complementation assays and an in vitro fluorescence-based transport assay, we confirm key residues required for MlaCD function. Finally, through molecular dynamics (MD) simulations, we provide insight into the trajectory of GPL transfer between MlaC and MlaD. Overall, by probing the MlaCD interaction we provide additional understanding of the intricate GPL transport mechanism via the Mla pathway.

## Results

### Stabilisation of the MlaCD[32-183] complex through cardiolipin binding

Initial efforts to obtain a stable MlaC-MlaD[32-183] complex by co-incubation failed. Purification by size exclusion chromatography yielded separate species suggesting low binding affinity (Fig. 1A - peaks 1 & 2 left panel and Fig. 1B). However, by first removing all lipid (as we have performed previously[9]) then incubating MlaD[32-183] with cardiolipin, we found that following co-incubation with MlaC-apo, the two proteins formed a stable complex (Fig. 1A - peak 2 right panel and Fig. 1B). Complexation was further validated by analytical ultracentrifugation, which showed a clear difference in the sedimentation coefficient

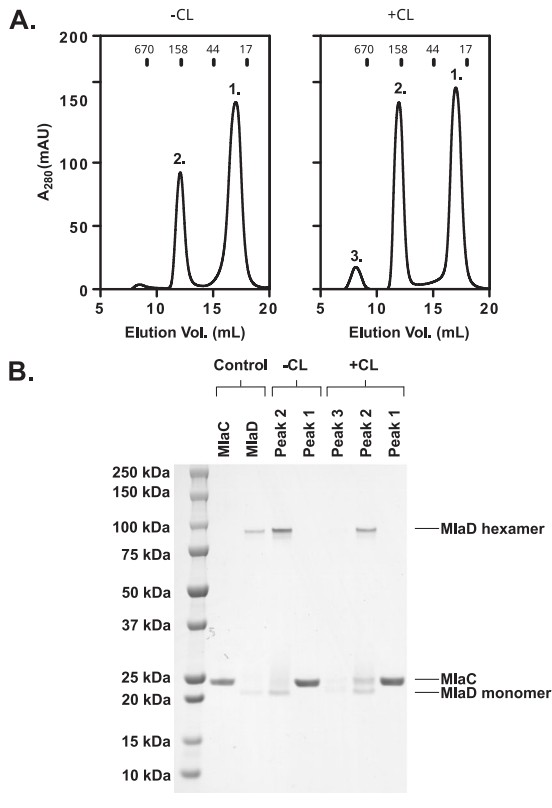

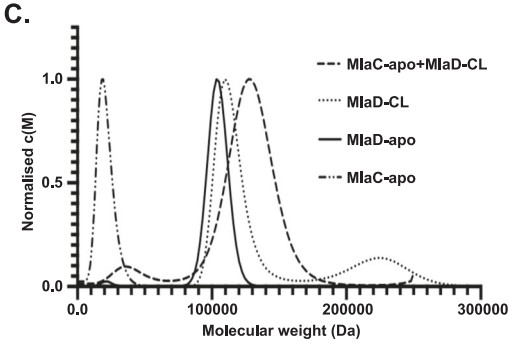

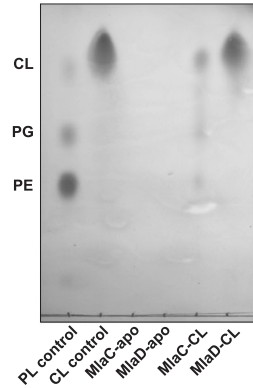

**Fig. 1 | Cardiolipin stabilises the MlaCD complex. A** Size exclusion chromatography (Superdex 200) analysis of MlaC and MlaD and their copurification in the absence (CL-) and presence (CL+) of cardiolipin. Peak numbers are indicated above the peaks. Peak 1 refers to MlaC elution, Peak 2 MlaD elution and Peak 3 aggregate. In the presence of cardiolipin, the ratio of Peak 1 to Peak 2 is altered showing increased protein in Peak 2. **B** SDS-PAGE of peak fractions from (**A**) showing the presence of MlaC co-purifying with MlaD in Peak 2 in the presence of cardiolipin only. Representative data of *n* > 3 independent purifications. **C** Analytical ultracentrifugation sedimentation velocity analysis of the effect of cardiolipin on MlaCD complex formation. **D** Thin layer chromatography showing the presence of cardiolipin within the MlaC fraction isolated from (**A**), samples are compared against a polar lipid (PL) control composed of CL, phosphatidylglycerol (PG) and phosphatidylethanolamine (PE). Source data are provided as a Source data file.

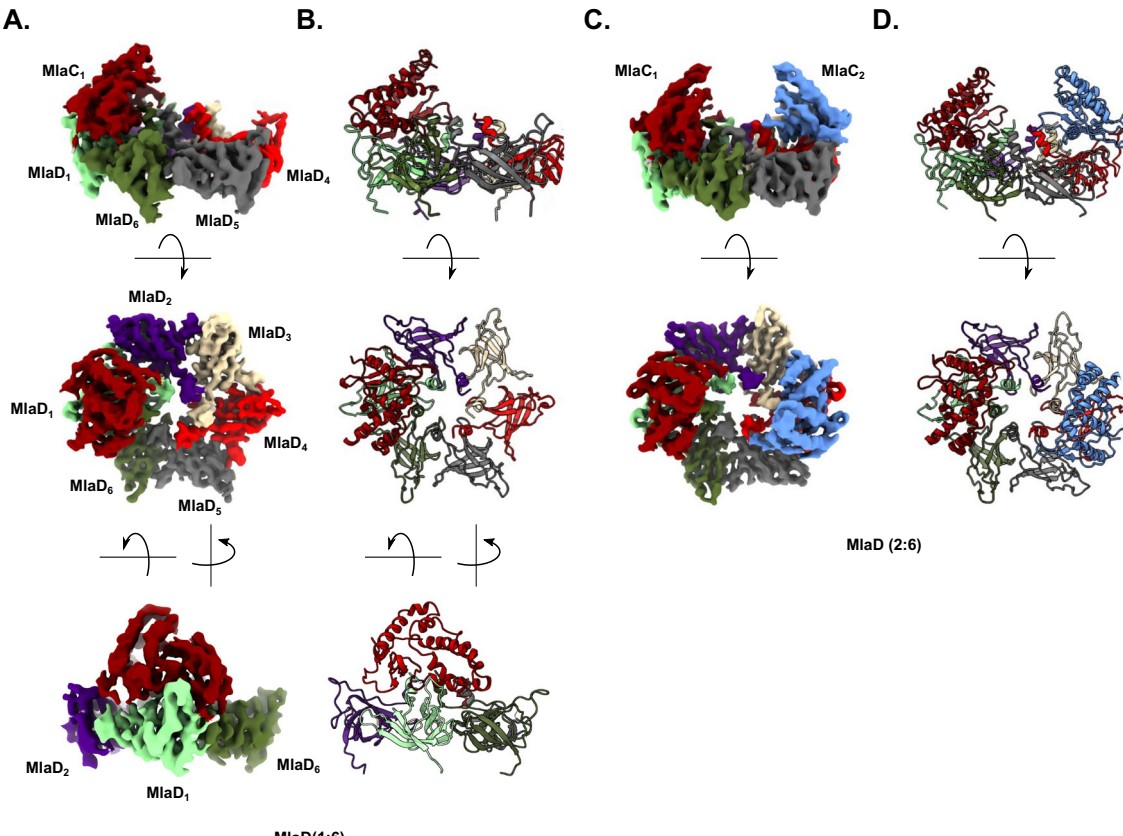

**Fig. 2 | Structure of the MlaCD complex, with two distinct stoichiometries.**
**A**, **C** Side view (top panel), top view (middle panel) and front view (bottom panel in A) of the cryo-EM map of MlaCD, with a 1:6 stoichiometry (**A**) or 2:6 stoichiometry (**C**). **B**, **D** Cartoon representation of the corresponding atomic models. Maps and atomic models are coloured by chain as follows: $MlaC_1$ (dark red), $MlaC_2$ (dark blue), $MlaD_1$ (light green), $MlaD_2$ (purple), $MlaD_3$ (beige), $MlaD_4$ (light red), $MlaD_5$ (grey) and $MlaD_6$ (dark green).

between the individual components (MlaC-apo, $MlaD^{32\text{-}183}$-apo & $MlaD^{32\text{-}183}$-CL) and the MlaC-apo:$MlaD^{32\text{-}183}$-CL complex (Fig. 1C). Our previous work, focusing on exchange of natively bound lipids did not observe this stabilisation[9], presumably because the concentration of cardiolipin was too low. Although stabilisation was observed, some MlaC was able to be separated from $MlaD^{32\text{-}183}$-CL and showed the presence of cardiolipin bound (Fig. 1D). However, compared to PG and PE exchanged between natively prepared $MlaD^{32\text{-}183}$ and MlaC-apo[9,19], the observed exchange was minimal. Furthermore, it remains unclear whether this was a true binding event with CL bound within the central cavity, or just loosely associated. Overall, the observed stabilisation of the MlaC-$MlaD^{32\text{-}183}$ complex (thereafter termed $MlaCD^{32\text{-}183}$) in the presence of CL and the decreased capacity for CL exchange, alongside an observed lack of CL bound MlaC in literature[19], suggests that within the cell CL is unlikely to be a natural substrate for the Mla pathway as this stabilisation likely impedes transport rates.

## The structure of the $MlaCD^{32\text{-}183}$ complex reveals two distinct stoichiometries

Next, we sought to determine the structure of the obtained $MlaCD^{32\text{-}183}$ complex, using cryo-EM. Initial screening indicated that the complex dissociated on EM grids at the concentrations utilised. To further stabilise the complex, we used glutaraldehyde cross-linking. A 30 s incubation with glutaraldehyde was sufficient to produce a stable complex, though with a molecular weight >250 kDa as estimated by SDS-PAGE (Supplementary Fig. 1A). Increased incubation with glutaraldehyde had little effect on the complex, suggesting that it was a stable species and not the result of aggregation. The glutaraldehyde

cross-linked sample was further purified by exclusion chromatography and used for structure determination (Supplementary Fig. 1B).

Using this cross-linked complex, we were able to produce cryo-EM grids suitable for structure determination and collected over 9000 micrographs. 2D class averages were feature-rich, and individual components were readily identifiable (Supplementary Fig. 2). We observed the complex form higher-order assemblies consisting of two back-to-back $MlaD^{32\text{-}183}$ hexamers, bound together via the MlaD surface facing MlaE within the MlaFEDB complex, explaining the molecular weight observed for the complex by SDS-PAGE (Supplementary Fig. 1A). This higher-order oligomerisation is clearly an artefact of glutaraldehyde cross-linking, not representative of any physiological interaction. Masked refinement and 3D classification, focusing on one MlaD hexamer, revealed two distinct species, one with a single MlaC bound to a $MlaD^{32\text{-}183}$ hexamer, and another with two MlaC molecules bound, with final resolutions of 4.35 Å and 4.38 Å, respectively (Fig. 2, Supplementary Table 1). The 3D classification of these two populations had approximately the same number of particles in each class, suggesting that they exist in equimolar equilibrium in solution.

The two structures are largely similar, with $MlaD^{32\text{-}183}$ forming the characteristic MCE hexamer. In the MlaCD(1:6) structure (Fig. 2A, B), a single copy of MlaC is bound offset from the central cavity, on top of the main MlaD ring, predominantly interacting with a single $MlaD^{32\text{-}183}$ monomer (henceforth termed $MlaD_1$) but making additional interactions between both the clockwise ($MlaD_2$) and counter-clockwise $MlaD^{32\text{-}183}$ ($MlaD_6$) subunits. In the MlaCD(2:6) structure (Fig. 2C, D), two MlaC molecules are bound on opposite sides of the $MlaD^{32\text{-}183}$ hexamer, with overall 2-fold symmetry.

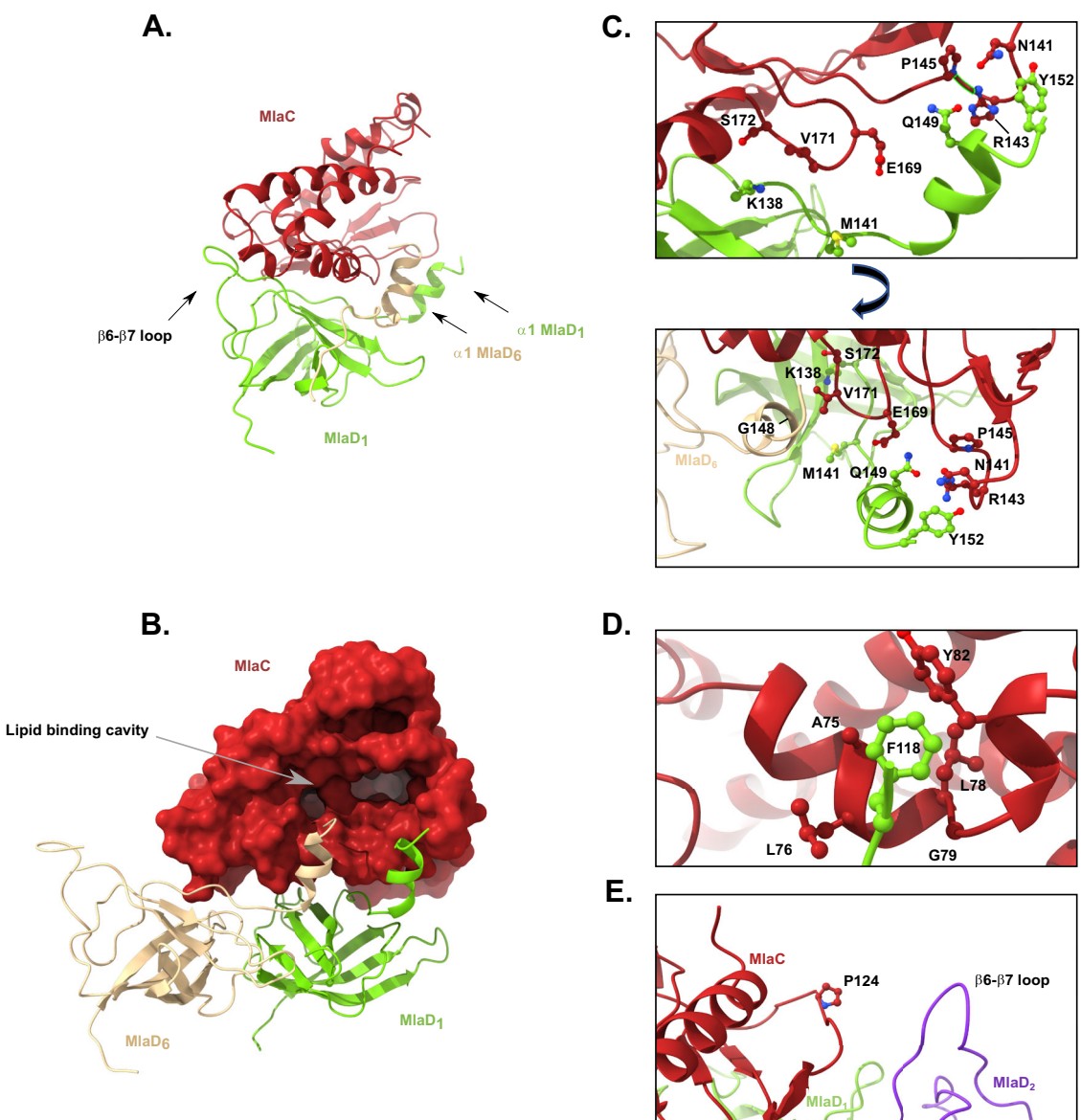

**Fig. 3 | The binding interface of MlaCD. A** MlaC (red) is shown bound to the MlaD hexamer from the side. MlaD$_1$ (green) and the α1 helix of MlaD$_6$ (beige) are shown highlighting how MlaC is pinched between the β6-β7 helix and the central helix assembly. Chains are coloured as in Fig. 2. **B** As in (**A**) but the MlaC surface is shown highlighting the position of the lipid binding cavity between the α1 helices of MlaD$_1$ and MlaD$_6$. **C** α1 helix interface residues identified through cross-linking[19] with likely interacting partners identified. Upper panel lacks MlaD$_6$ for clarity. **D** β6-β7 loop interface residue F118 identified through cross-linking with likely interacting partners identified. β6-β7 loop residues 120–122 cut away for clarity. **E** Pro124 identified through cross-linking showing the absence of close contacts to MlaD$_2$ (purple) in the MlaCD(1:6) structure presented here.

Comparison between the MlaCD(1:6) and MlaCD(2:6) structures (focused on comparing the Cα's of each bound half of the complex e.g. MlaC$_1$:MlaD$_{1,2,6}$ in MlaCD(1:6) with MlaC$_1$:MlaD$_{1,2,6}$ and MlaC$_2$:MlaD$_{3,4,5}$ in MlaCD(2:6)) resulted in an observed RMSD of ~2.0 Å to both halves suggesting there are no major differences in conformation between the structures.

Further 2D and 3D classification of the data did not reveal any evidence for particles with additional copies of MlaC bound to the MlaD$^{32-183}$ hexamer, or for any MlaCD(2:6) structures with the two MlaC monomers bound in any orientation other than directly opposite each other. However, it is yet to be resolved whether 1 or 2 MlaC monomers per MlaD hexamer is the native biological assembly. Data presented by MacRae et al. suggest that orientations where MlaC binds to residues that are not directly opposite do exist[20], however, this may be an artefact of their methodology of stabilising the interaction by increasing avidity with a trimerised MlaC construct to promote multivalent binding. Our

structure, supplemented by the prior cross-linking study carried out by Ercan et al. suggests that there is involvement of the MlaD monomers adjacent to the primary binding monomer as shown in Fig. 3, which, along with direct steric hindrance between MlaC molecules, would likely interfere with binding of MlaC at these loci[19]. MacRae et al. also suggested that binding would be unlikely to occur at the MlaD$_2$ or MlaD$_6$ positions, however, their data suggests that binding at the MlaD$_3$ position is possible[20]. While, we agree that there would not be steric hindrance between the binding MlaC molecules in this arrangement, the lack of evidence for such a binding orientation in our data combined with several other considerations, such as the breakdown of the 2-fold symmetry of the MlaFEDB complex, the necessity for dual involvement of the MlaD$_2$ monomer in two binding events, and the nature of the structures presented by MacRae et al., which appears to be more of a transient docking of MlaC to the β6-β7 loop rather than a binding event permitting functional lipid exchange lead us to conclude that the

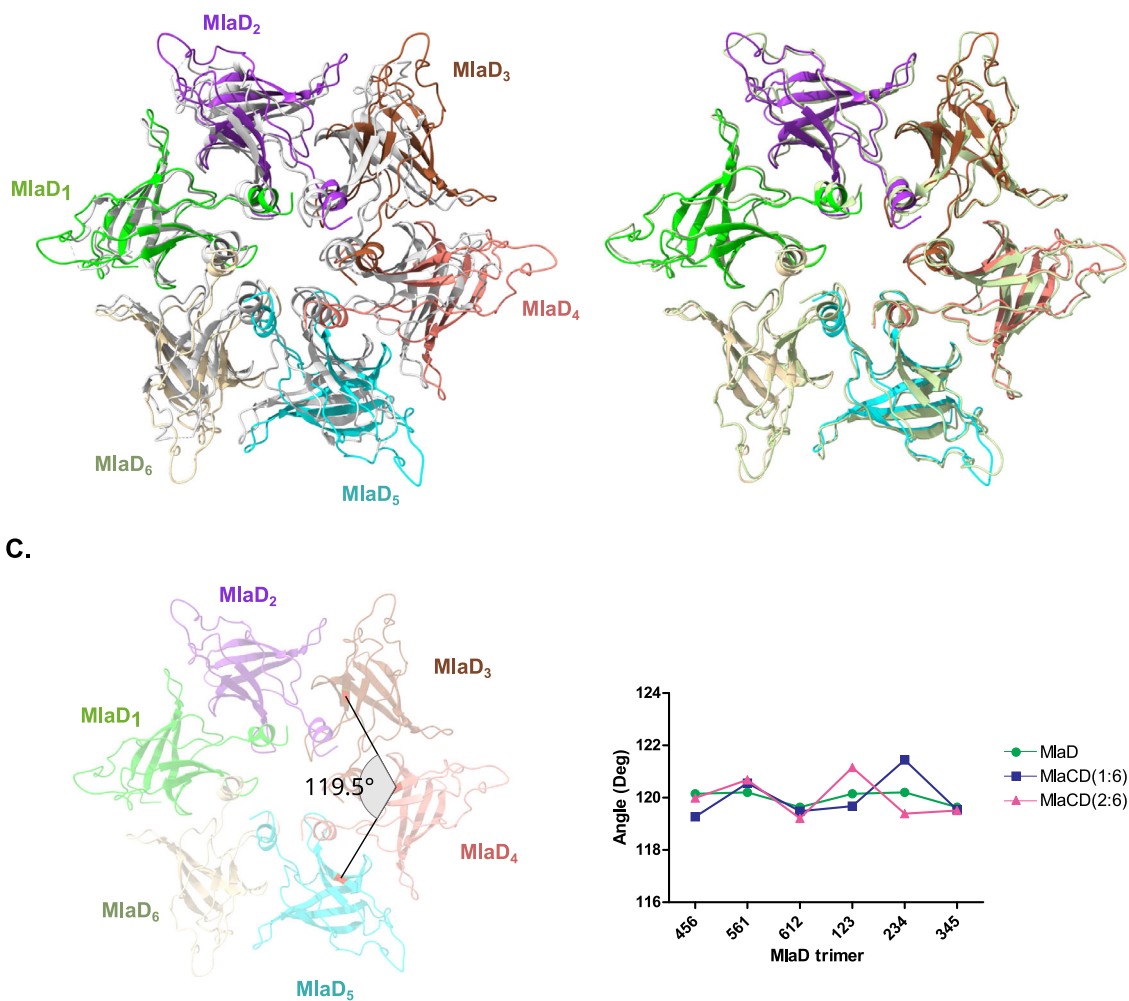

**Fig. 4 | Structural changes to MlaD upon binding to MlaC. A** Overlay of MlaCD(1:6) (coloured from MlaD₁ to MlaD₆ in green, purple, brown, pink, cyan and beige respectively) with the MlaD hexamer from the apo crystal structure. C2 symmetry axis shown (5UW2, grey). The hexamer is aligned to the MlaD₁ chain. In the structure bound to MlaC, the hexamer no longer adopts a strict 6-fold symmetry, with a notable enlargement of the structure. **B** Overlay of MlaCD(1:6) with MlaD(2:6) coloured according to (**A**). **C** Angle changes observed between MlaD monomers between the apo structure (5UW2) (green), MlaD(1:6) (blue) and MlaD(2:6) (pink). Angles were measured between MlaD trimers by measuring the angle between an arbitrarily picked residue (113) in each monomer (red) as shown. Source data are provided as a Source data file.

binding orientations we have observed are the only orientations that permit lipid exchange in vivo.

**Structural changes to MlaD³²⁻¹⁸³ observed in the MlaCD complex**
Within the limitations of the map's resolution, pairwise comparison between MlaD protomers within the MlaD hexamer suggest that all six copies of the protein adopt similar conformations in the core fold (residues 40–141). MlaCD(1:6) has a Cα RMSD of between 0.7 and 0.8 Å in the core fold across all pairwise protomer alignments. Likewise, MlaCD(2:6) has a Cα RMSD of between 0.7 and 0.9 Å in the core fold across all pairwise protomer alignments, except for those positioned directly opposite in the hexamer, which have RMSDs as low as 0.3 Å in the core fold and 0.4 Å for the entire Cα trace. The MlaD₁,₄ and the MlaD₃,₆ pairs have significantly lower RMSDs for the full Cα trace, likely owing to the interaction with MlaC restricting positional variability in the central helix and β6-β7 loop. In contrast, pairwise comparison of the adjacent MlaD₁,₆ pairs in both the MlaCD(1:6) and MlaCD(2:6) structures and the MlaD₃,₄ pair in the MlaCD(2:6)

structure have an RMSD of 1.8 Å for the full Cα trace owing to differential interactions of the central helices with MlaC resulting in them moving apart, creating somewhat of a gap between the helices of these monomers. RMSDs for the pairwise comparison of the total Cα trace for the remaining MlaD pairs falls in the range of 0.9–1.2 Å for the MlaCD(1:6) structure and 1.3–1.5 Å for the MlaCD(2:6) structure.

Similarly, comparison with the previously published crystal structure of MlaD³²⁻¹⁸³ in isolation (5uw2) shows that each MlaD monomer of MlaD³²⁻¹⁸³ undergoes little conformational change upon binding to MlaC, with each monomer having an overall Cα RMSD of 1.3–1.4 Å between the apo- and MlaC-bound states, with the major structural deviations being in the central helix and β6-β7 loop. Although conformational changes between the core fold of monomers were minimal, comparison of the MlaD hexameric architecture between the apo- and MlaC-bound states revealed a striking rearrangement of the MlaD³²⁻¹⁸³ hexamer, with a clear expansion of the ring observed (Fig. 4A). Comparisons between MlaCD(1:6) and MlaCD(2:6)

show this expansion occurs on binding the first molecule of MlaC, with few changes in the structure of the hexamer noted between MlaCD1:6 and 2:6 (Fig. 4B). In the case of MlaCD(1:6) binding clearly results in the 6-fold symmetry of the MlaD hexamer being broken, as evidenced by changes in the angles between MlaD monomers (Fig. 4C). Interestingly, the largest deviations occur distal from MlaC binding, for example between MlaD monomers 2, 3 and 4. Importantly, such a break in symmetry was not observed in the structures of MlaD within the MlaFEDB inner-membrane assembly[12–17], and is therefore possibly not representative of the physiological binding event. While we attribute the expansion of the MlaD ring and reorganisation of the helices to the MlaC binding event, it is equally possible that the ring expansion is due to accommodate the CL lipid, or a result of chemical cross-linking and the formation of the non-physiological dodecameric MlaD complex. MD simulations of the interactions between MlaC and MlaD in the presence of lipids seem to suggest that the binding of MlaC does not cause reorganisation of the MlaD central helix, however, the interaction between MlaD and lipids alone does cause significant reorganisation (Supplementary Video 3). Although, as we were unable to resolve any lipids in our structural data we cannot confirm if the reorganisation we observe in the MlaCD (1:6) and (2:6) structures is due to the passage of lipids between the helices.

## Structural basis for the interaction between MlaC and MlaD

In the structures described above, MlaC appears pinched between the central helical bundle and the rear β6-β7 loop of MlaD$_1$ (Fig. 3A), such that its lipid binding pocket is directed towards the interface between α1 of MlaD$_6$ and α1 of MlaD$_1$ (Fig. 3B). We note that although purified in the presence of CL to stabilise the complex (see above), no lipid density was observed within the previously reported lipid binding pockets of MlaC or MlaD.

The interaction interface itself is made up of predominantly electrostatic interactions, with MlaD$^{32-183}$ forming a largely electro-negative surface, composed of residues from β5, β6, β8 and α1 together with the β6-β7 loop (Supplementary Fig. 3A), whilst MlaC features a positively charged groove that complements the negatively charged β6-β7 loop of MlaD (Supplementary Fig. 3B, C), composed of residues found within helices α3, α4, α6, the β4-α6 loop and β2-β3 loop.

The structure is consistent with co-evolution analysis, with numerous regions of co-evolution observed at the interface between MlaC and MlaD (Supplementary Fig. 4 and Supplementary Table 2) and previous pBpa cross-linking which identified Pro124, Glu169, Val171 and Ser172 in MlaC and Phe118, Met141, Gln149 and Tyr152 in MlaD as involved in the MlaCD interaction[19]. MlaC Glu169, Val171 and Ser172 together with MlaD Met141, Glu149 and Tyr152 all cluster around the interaction of loops β2-β3 and β4-α6 in MlaC and the central α1 helix bundle (Fig. 3C). At the resolution observed, specific information on side chain interactions is lacking but it appears that MlaC Glu169 interacts with α1 of MlaD$_1$. MlaC Val171 interacts with α1 of MlaD$_6$ and appears to sit within a small pocket created by Gly148. Whilst Ser172 is most closely aligned to Lys138 of MlaD$_1$. Finally, Met141 in MlaD is most closely aligned to MlaC Gly170, and Gln149 and Tyr152 with residues Asn141-Pro145 of MlaC loop β2-β3.

Of the remaining residues identified through pBpa cross-linking, MlaD Phe118 sits within the β6-β7 loop and is in close proximity to a hydrophobic patch on MlaC composed of residues Ala75, Leu78-Tyr82 (Fig. 3D). Whilst MlaC Pro124, does not make any observed close contacts in our structure, however, based on its position, the most likely location for interaction would be with the β6-β7 loop of MlaD$_2$ (Fig. 3E).

Critically, we emphasise that our structure is consistent with previously reported pBpa cross-linking obtained in bacterial cells. This suggests that we have likely captured the true complex, which is not majorly affected by using a sub-complex and/or chemical cross-linking.

## The β6-β7 loop is essential for function

As indicated above, the interaction of the MlaD β6-β7 loop with the distal end of the MlaC β-sheet suggests a potential role for this loop in the function of the complex. To assess this, we first performed a series of complementation studies. Using a plasmid encoded copy of full-length MlaD (including its N-terminal helix anchor) within a ΔmlaD background, we assessed the role of residues within this loop: the ΔmlaD mutant is unable to support growth on an SDS/EDTA background, but growth can be rescued by ectopic expression of MlaD (Fig. 5A), allowing us to screen for a range of mutants.

Central to providing the negative charge of the MlaD β6-β7 loop are residues E119, D120 and E122 (Fig. 5B). Charge reversal of either E119 or D120 to the bulkier amino acid lysine resulted in an inability to grow on SDS/EDTA (Fig. 5A). In contrast, charge reversal for residue E122 (E122K) had no impact on growth. Next, we assessed the role of F118, this hydrophobic residue also resides within the β6-β7 loop. Here, we investigated whether its hydrophobicity had a role in function by exchanging it to a glutamate, this also was completely disruptive to growth on SDS/EDTA.

To confirm that these mutations were causing disruption to function rather than destabilising the protein fold, attempts were made to assess MlaD levels within the cell. Unfortunately, we were unable to generate an MlaD antibody sufficiently sensitive to assess native MlaD levels. Therefore, to give some indication whether the mutations affected stability, all mutants (including subsequent MlaD mutants discussed) were over-expressed and purified. All produced stable proteins that were used in subsequent assays (Supplementary Figs. 5 and 6), suggesting the mutant phenotypes observed were unlikely the result of protein destabilisation.

Next, we sought to more directly relate the growth defects caused by these mutations to a reduced capacity for lipid exchange in the Mla pathway. To do this we employed a FRET-based PL transport assay described by Tang et al. incorporating the complete Mla pathway[15]. Readout of transport is reported by the reduction of quenching of NBD labelled phosphatidyl glycerol (PE) by Rhodamine-PE after transport from donor MlaA containing proteoliposomes to acceptor MlaFEDB proteoliposomes by the MlaC chaperone.

For the positive control with WT protein, we observed an initial acceleration in fluorescence increase before it began to plateau (Fig. 5C). The negative control was performed with the system lacking MlaFEDB in the acceptor liposome and showed a significant attenuation in signal, associated with a lack of lipid transport. Subsequent experiments were assessed relative to the activity observed by these controls.

Analysis of MlaD F118K, E119K and D120K reveal that they all show significantly reduced GPL transport compared to the WT (Fig. 5C), with F118K showing transport comparable to the MlaFEDB negative control. The E119K and D120K mutants showed a capacity for lipid exchange marginally above the level of the negative control, but both had approximately 5-fold less activity than the WT. In contrast, the E122K mutant showed no significant deviation in activity when compared to the WT. Lipid transport assays for all MlaD mutations were in line with their ability to support growth on SDS/EDTA.

Next, we investigated whether reciprocal mutations within MlaC also impacted function. Mutations within the binding groove for the β6-β7 loop, namely Y72F, L76R and Q80E were therefore investigated. All plasmids could be assessed for leaky expression by western blot analysis (Fig. 5A), confirming all produced stably expressed protein. Both Y72F and L76R showed minimal perturbation to growth on SDS/EDTA with L76R showing a slightly stronger phenotype (Fig. 5A). Y72F had no effect on reducing lipid exchange, but L76R did exhibit marginal reduction compared to the WT (Fig. 5D). However, the Q80E mutation was completely prohibitive to growth on SDS/EDTA and reduced lipid transport comparative to the E119K and D120K MlaD mutants.

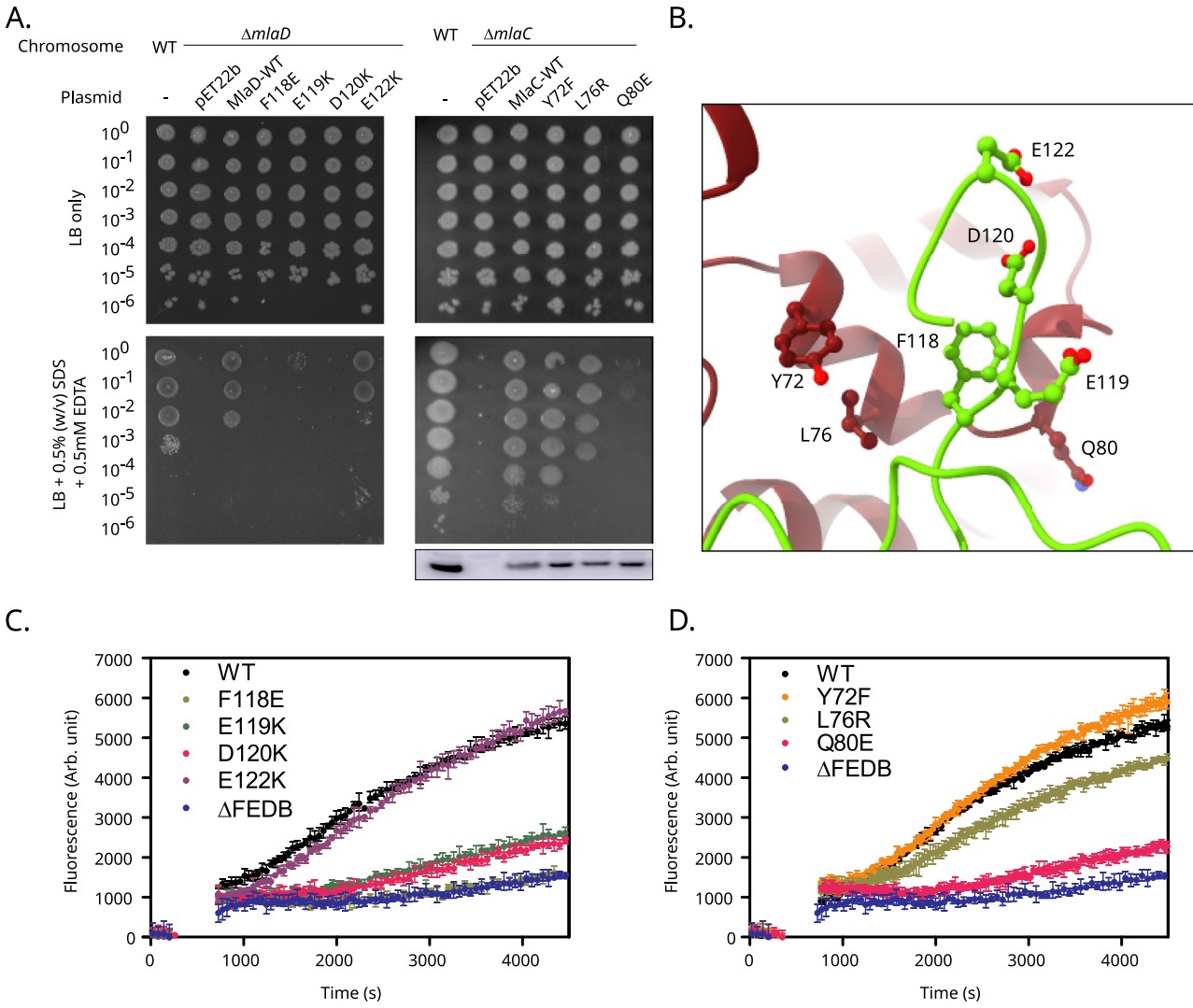

**Fig. 5 | The β6-β7 loop is essential for function. A** Screen for SDS/EDTA sensitivity of cells carrying pET22b encoding the WT or mutated copies of MlaC or MlaD in the parent or *ΔmlaC/ΔmlaD* strain background. WT – BW25113 parent strain. Cells were normalised to an $OD_{600}$ of 1 and 10-fold serially diluted before being spotted on LB agar containing the indicated condition. Western blot showing levels of MlaC within the cell. Full blot presented in Supplementary Fig. 8. **B** The positioning of the β6-β7 loop (green) interacting with MlaC (red) is shown, highlighting the residues mutated in (**A**). **C, D** FRET-based GPL transport assays. Fluorescence increase corresponds to a reduction in NBD-PE FRET quenching by Rhodamine-PE as lipids are transferred from the MlaA proteoliposome to the MlaFEDB proteoliposome.

Excitation wavelength 460 nm, emission wavelength 535 nm. Representative data from *n* = 3 independent experiments. Data in (**C**) and (**D**) show the mean and range from triplicate experiments. The traces in (**C**) correspond to the F118E (olive green), E119K (green), D120K (pink) and E122K (purple) MlaD mutants investigated in this study alongside the positive WT control (black) and the ΔMlaFEDB negative (blue). The traces in (**D**) correspond to the Y72F (orange), L76R (olive green) and Q80E (pink) MlaC mutants, which are proximal to the β6-β7 loop of MlaD during inter-action, alongside the positive WT control (black) and the ΔMlaFEDB negative (blue). Source data are provided as a Source data file.

Overall, these results suggest the MlaD β6-β7 loop interaction with MlaC observed within the cryo-EM structure is functionally rele-vant as several mutants are capable of severely attenuating or com-pletely abolishing lipid transport. While it is not clear exactly which residues are interacting between the two proteins due to the resolu-tion limit of our structural data, it is evident that both charged-based electrostatic interactions and the hydrophobic interactions of non-polar residues are involved in the functional docking of the β6-β7 loop.

**Access between the α1 helices is important for activity**

To investigate the significance of the observed break in the MlaD hexamer symmetry in the presence of MlaC (see above, Fig. 4), we created a series of cysteine mutants: two single mutants, Q149C and L151C, as well as the double mutant Q149C:L151C. The Q149C:L151C mutant was designed with the intention of inducing disulphide bond formation between all adjacent α1 helices in the hexameric assembly,

completely rigidifying the central pore (Fig. 6A). SDS-PAGE of the purified Q149C:L151C mutant under reducing/non-reducing condi-tions confirmed the disulphide bond formation occurs between monomers, stabilising the hexamer (Fig. 6B). Growth of L151C and Q149C:L151C on SDS/EDTA was completely prohibitive, however, the growth of Q149C was comparable to the rescue plasmid (Fig. 6C). Growth of the rescue plasmid on SDS/EDTA in a reducing background (2 mM TCEP) could not be supported and even the WT BW25113 was severely affected under these conditions indicating that TCEP together with SDS/EDTA too severely compromised cell growth and could not be used to further investigate these mutations. Thus, the recovery of function in reducing conditions was assessed using the GPL transport assay already discussed. Assessment of GPL transport capabilities under non-reducing conditions suggested that both single mutations individually were prohibitive to activity, as was the double mutation (Fig. 6E). The activities of both single mutants were recovered by the

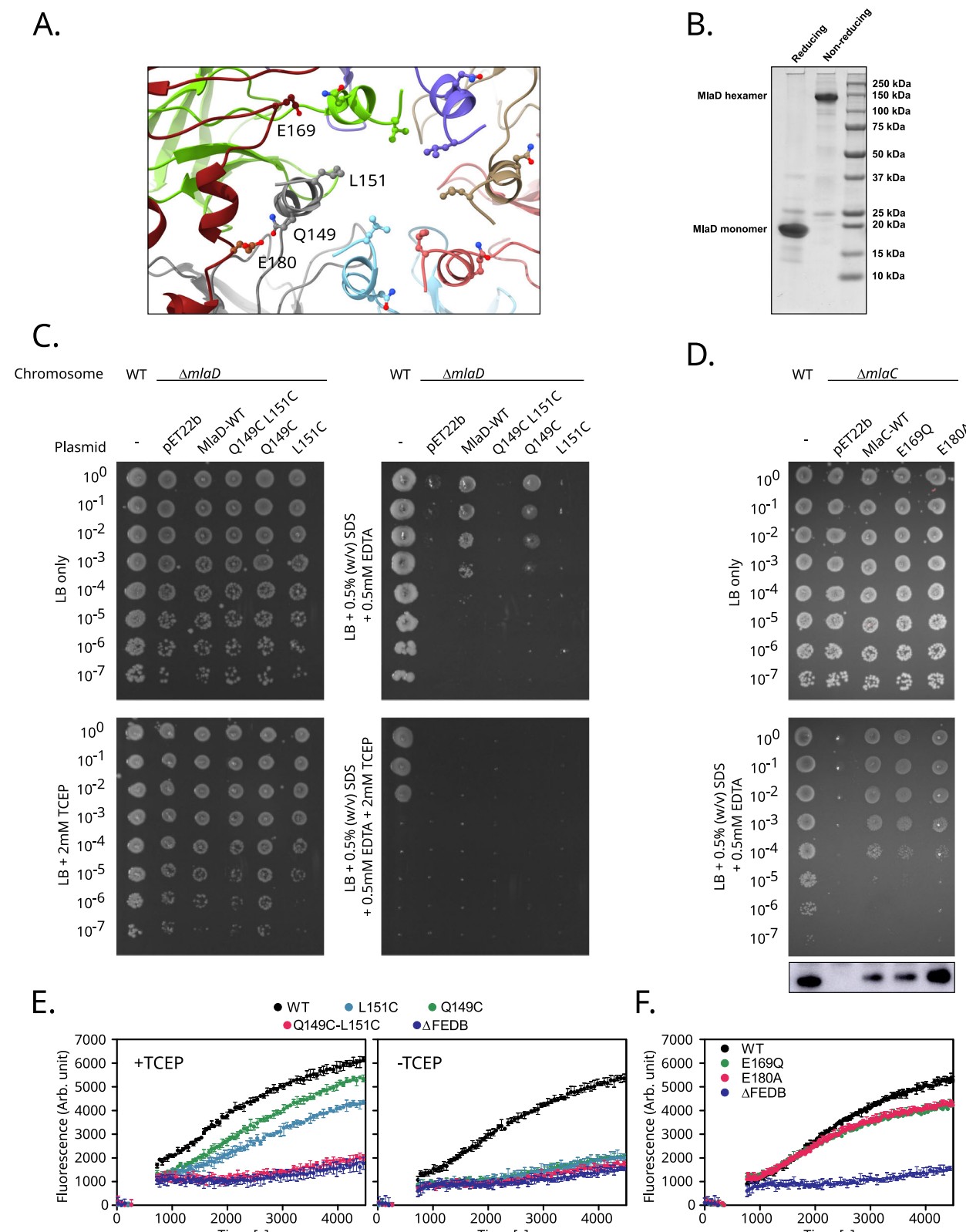

addition of TCEP, the double mutant however, was still incapable of GPL transport even in reducing conditions. The results observed from the single cysteine mutants would suggest that disulphide bond formation occurred in the absence of TCEP, dimerising adjacent monomers (forming 3 disulphide bonded dimers in the hexamer), which was sufficient to impede transport. Under reducing conditions, the observed activity suggests that the single cysteine substitutions alone do not affect transport. Why the Q149C:L151C double mutant remains inactive under TCEP conditions remains unclear, but it is likely just the result of an intolerance to multiple mutations to the helical interface potentially impacting MlaC binding or modifying the charge/hydrophobicity sufficiently to prevent lipid passage.

Finally, as it was noted that the Q149C mutant is not prohibitive to growth on SDS/EDTA, but does prevent GPL transport in vitro, we

**Fig. 6 | Access between the α1 helices is important for activity. A** Focus looking down the central channel of MlaD. Positions of residues mutated highlighted. MlaC coloured red, MlaD coloured according to Fig. 4. **B** SDS-PAGE of MlaD Q149C:L151C boiled under reducing and non-reducing conditions confirming disulphide bond formation occurs between monomers stabilising the hexameric form. **C** Screen for SDS/EDTA sensitivity of cells carrying pET22b encoding the WT or mutated copies of MlaD in the parent or ΔmlaD strain background. WT – *E. coli* K12 BW25113 parent strain. Cells were normalised to an $OD_{600}$ of 1 and 10-fold serially diluted before being spotted on LB agar containing the indicated condition. **D** Screen for SDS/EDTA sensitivity of cells carrying pET22b encoding the WT or mutated copies of MlaC in the parent or ΔmlaC strain background. Corresponding western blot showing levels of MlaC within the cell. Full blot presented in Supplementary Fig. 8.

**E**, **F** FRET-based GPL transport assays. Fluorescence increase corresponds to a reduction in NBD-PE FRET quenching by Rhodamine-PE as lipids are transferred from the MlaA proteoliposome to the MlaFEDB proteoliposome. Excitation wavelength 460 nm, emission wavelength 535 nm. Representative data from $n = 3$ independent experiments. Data in E and F show the mean and range from triplicate experiments. The traces in E) correspond to the L151C (light blue), Q149C (green) and Q149C:L151C (pink) cysteine mutants of MlaD alongside the WT (black) and ΔMlaFEDB negative (blue) controls. The traces in F) correspond to the E169Q (green) and E180A (pink) mutants of MlaC, which are proximal to the MlaD α1 helix during interaction, alongside the positive WT (black) and ΔMlaFEDB negative (blue) controls. Source data are provided as a Source data file.

propose that the observed effect may be dependent on the differential properties of unbonded cysteine residues under the relatively oxidative conditions in the periplasm compared to the conditions of the in vitro assay.

To further assess the role of the α1 helix, we performed mutations in MlaC to assess whether they were able to disrupt function (Fig. 6A, D, F). Initially we investigated E169, which interacts with α1 of MlaD1 proximal to Q149 and is potentially implicated in maintaining the position of the MlaC β4-β5 loop. This was mutated to glutamine to determine if charge interactions had any role to play in this interaction. This showed no disruption to growth or GPL transport (Fig. 6F). Similarly, E180, which interacts with $MlaD_6$, when mutated to alanine (E180A) also showed no effects on growth and GPL transport (Fig. 6D, F). The tolerance to these single amino acid substitutions suggests a lack of defined interaction between residues between the α1 helix and MlaC. This is backed up by a lack of co-evolution within this region (Supplementary Table 2).

### Molecular dynamics of lipid transfer between MlaC and MlaD

To provide additional insight into the dynamics of binding and lipid exchange between MlaC and MlaD in the absence, and then subsequently the presence of the MlaFEB complex, we conducted a series of MD simulations investigating the movement of lipid within the stabilised MlaCD complex we had generated (Supplementary Videos 1, 2 and 3). While this does not provide direct insight into the mechanism of function of the MlaFEDB complex it does assist in understanding observations made when investigating the function of lipid exchange between MlaC and soluble MlaD.

Initially, we investigated how the MlaC:MlaD complex interacts with lipids within a GPL bilayer. Using alphafold[21] we first modelled in the N-terminal transmembrane helices onto the $MlaCD^{32-183}$ complex, to more accurately represent the complex within the membrane, then performed 5 μs simulations of MlaCD within a model lipid bilayer (Fig. 7). Up to four lipids were observed to move out of the GPL bilayer into the central cavity of MlaD through initial transfer of a single acyl tail (Fig. 7A). However, no movement of GPL to MlaC was observed. The binding of a maximum of 4 lipids is consistent with the observations of Thong et al. as well as our own observations with MlaD alone (Supplementary Video 3) and suggests the binding of MlaC does not impact the number of lipids MlaD can accommodate[10]. While these observations do not inform the default binding state of MlaD as part of the MlaFEDB complex, they do give insight into the dynamics of the bound state observed by Thong et al. and inform us on the maximal binding capacity of MlaD.

Next, we sought to investigate GPL binding between MlaC:MlaD in more detail. By investigating GPL interactions in the presence of free lipids (Mix of PE & PG randomly placed within the experimental frame), we observed the stable binding of a single PG to MlaC in both MlaCD 1:6 and 2:6 stoichiometries (Fig. 7B, C). Furthermore, during repeat simulations we also observed the occasional simultaneous binding of a single PE GPL between both MlaC and MlaD, each bound via a single acyl tail (Fig. 8). Binding was between the α1 helices of MlaD1 and

$MlaD_6$, displacing them from their canonical position lining the central MlaD helix bundle. This binding is consistent with our earlier observations and is suggestive of an intermediate state during transport between MlaC and MlaD whereby lipids pass between the helices of the MlaD α1 helix bundle rather than over the top. This is further supported by multiple cryo-EM structures which show density associated with lipids/detergents partially intercalated between the helices in this region[14,16].

To understand if the same process occurs within the MlaFEDB complex we simulated binding of free GPLs to the MlaC:MlaFEDB complex. Here the MlaCD(1:6) complex structure was mapped in place of MlaD within PBD 7CGE. We also observed simultaneous binding of GPLs between MlaC and MlaD (Supplementary Fig. 7A, B). Furthermore, it was witnessed that over some simulations, binding between MlaC:MlaD occurred first followed by both acyl chains binding to MlaC (Supplementary Fig. 7B, C). In summary, all observed cases of lipid transport occurred through the sequential exchange of a single acyl tail.

## Discussion

Although much is now understood about the biogenesis mechanisms of the outer membrane, we are still largely in the dark regarding the transport of GPLs. Indeed, we have known that GPLs can move bidirectionally between the inner and out membranes for over 40 years[22–24]. Yet to date only the Mla retrograde GPL transport pathway has been confirmed biochemically as a GPL transporter, with much still to be resolved regarding its mechanism. Here, we provide insight into how the MlaC periplasmic GPL shuttle docks with MlaD by determining the structure of the $MlaCD^{32-183}$ complex, revealing that the MlaD hexamer is bound either with one or two monomers of MlaC, on opposing faces of the $MlaCD^{32-183}$ hexamer. It remains to be seen which conformation predominates within the cell. The binding interface is largely consistent with the deep computational modelling adopted by MacRae et al. to predict the structure of the MlaCD interaction[20], but unlike their low-resolution MlaC-MlaFEDB cryoEM models using a modified trimeric MlaC-T4 fibrin construct to increase avidity, we did not see any binding orientations other than two MlaC monomers bound directly opposite each other on the MlaD hexamer. We believe our complex structure adopts a more realistic native-like arrangement than that obtained by MacRae et al. who used a multidentate MlaC ligand which likely promoted a non-physiological binding arrangement with MlaC molecules bound at positions $MlaD_1$ and $MlaD_3$, rather than the $MlaD_1$ and $MlaD_4$ directly opposite orientation we observed. The MlaCD(2:6) structure is also consistent with the observed two-fold symmetry for the MlaFEB components[12–17]. Furthermore, the repeated observation of 2 GPLs within the MlaFEDB central binding pocket[12,13,17] and multiple structures of MlaC isolated in the presence of a single GPL, further supports this model, with MlaFEDB transporting multiple GPLs each transport cycle, one each to each MlaC monomer. However, many ABC transporters have been shown to have an asymmetric mechanism[25], suggesting the binding of a single MlaC monomer could also be feasible. The presence of two GPLs within the central binding

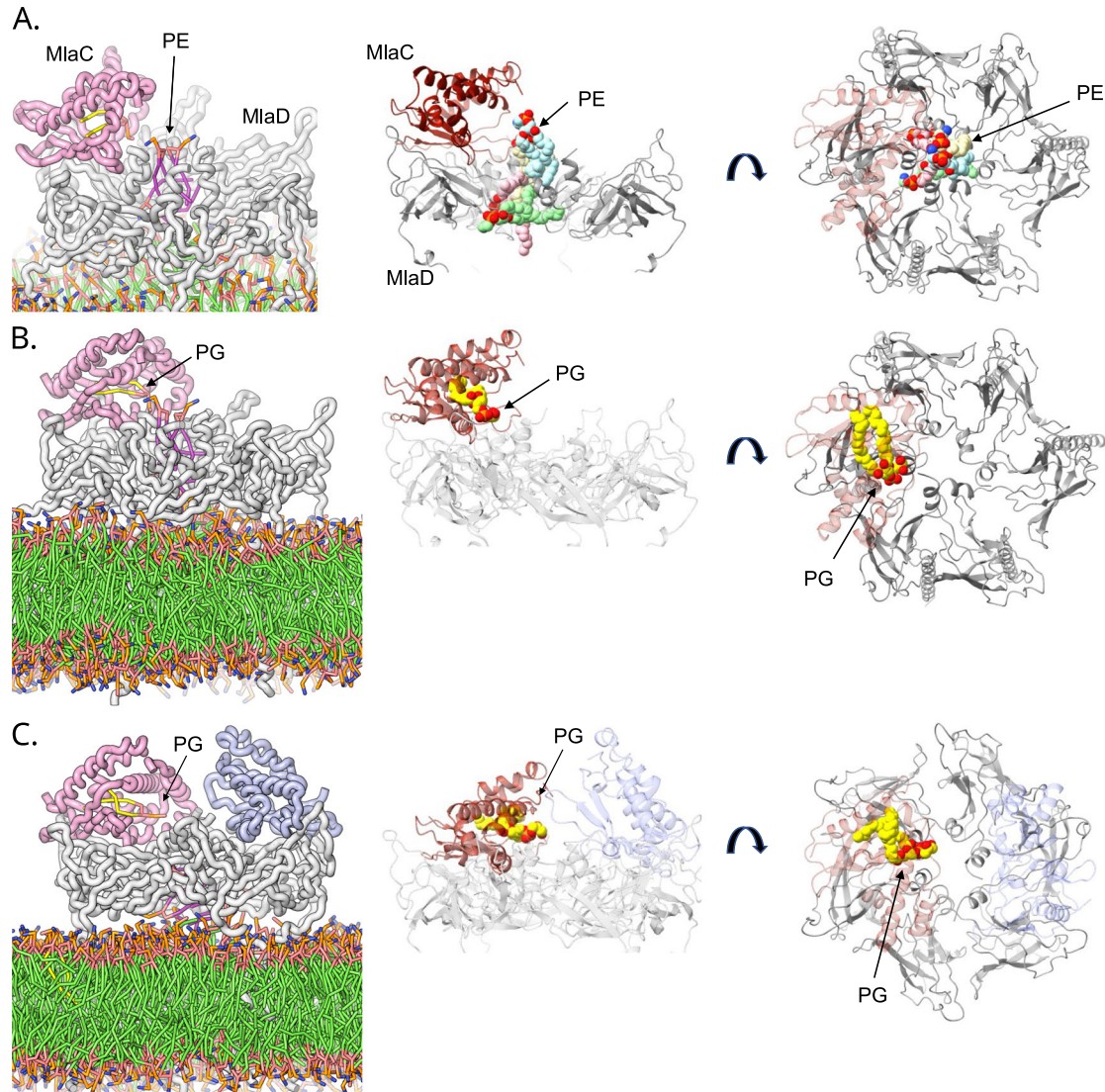

**Fig. 7 | Binding of lipids to membrane-bound MlaCD complex during 5 μs coarse-grained MD simulations.** The first images in rows (**A**) to (**C**) are depictions taken from Supplementary Videos 1, 2 & 3, respectively. These include the membrane and show the lipids in the binding pockets. The subsequent images show ribbon diagrams emphasising the position and orientation of the lipids. Both side and top views are presented. **A** Example snapshot of MlaCD (1:6) (maroon:grey) at 500 ns of the 5 μs simulation, showing 4x PE (2 of which; green and beige, bound through the bottom of the central MlaD pore and originated from the membrane and 2 of which; blue and pink, bound through the top of the pore and originated as free lipids in solution). **B** Focus on the 1x PG bound to MlaC in MlaCD (1:6). **C** Focus on the 1x PG bound to MlaC in MlaCD (2:6). Transmembrane helices modelled using AlphaFold[21]. CG simulations converted to AT representation using CG2AT.

pocket and a single MlaC bound can be reconciled by both GPLs binding to a single MlaC monomer. Although most MlaC structures solved to date show density for a single GPL, structures of MlaC homologues from both *P. aeroginosa* (pdb:6HSY) and *P. Putida* (pdb:4UWB) have been determined showing density for four acyl chains, suggesting binding of two GPLs within the pocket, though this could just be a single CL.

Through MD simulations, mutagenesis and FRET-based assays, we have shown that GPLs are transported between the α1 helices of the central MlaD pore, with MD simulations providing evidence that transport may occur via a single acyl chain at a time. This is consistent with observed structures of MlaFEDB in which GPLs are observed to adopt extended conformations[12,13,17], with one fatty acid tail directed down into the pocket of MlaE while the other extend up into the hydrophobic pore of MlaD.

Finally, we have identified the MlaD β6-β7 loop region as a significant site of interaction between MlaC and MlaD, with mutations within this loop, which complexes at the back of MlaC distal from the

MlaC binding site, completely negating GPL transport and function of the complex in vivo. This same region was predicted using alphafold modelling by MacRae et al., they tested the effect of mutation on binding affinity and observed mutations within this interaction site to significantly impact binding affinity between MlaC and MlaD[20]. The position of this loop interaction, at the back of MlaC, close to the pivot of the β-sheet, alludes to a mechanism by which it may control the opening/closing of the MlaC GPL binding cavity. First consider antero-grade transport observed by us and others[4,9,15]. We have shown here and previously that MlaC-apo is unable to take up GPLs from the environment. However, binding to MlaD, between the β6-β7 loop and the main body could pull the GPL binding cavity open allowing GPLs to enter. This is consistent with our structures of MlaCD, which show MlaC in an open configuration bound to MlaD. In the case of retro-grade transport, a conformational change in MlaD, as a result of ATP binding, could lead to either the upwards movement of β6-β7 whilst the main body of MlaD remains fixed, or vice versa, leading to the closing of the MlaC cavity and the expulsion of GPLs contained within.

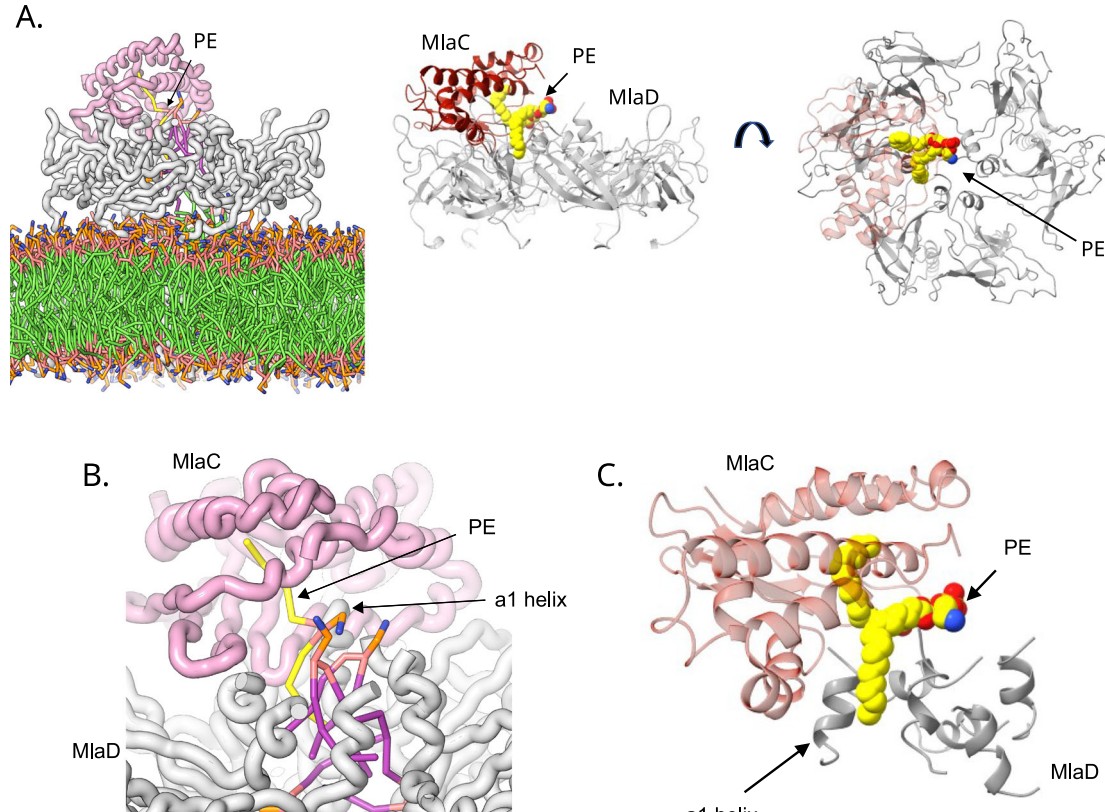

**Fig. 8 | Binding of a single PE lipid simultaneously between MlaC and MlaD during 5 μs coarse-grained MD simulations. A** Example snapshot of MlaCD (1:6) (maroon:grey) at 500 ns of the 5 μs simulation, showing PE binding simultaneously to MlaC and MlaD, each via a single tail, between the α1 helices of MlaD. The first image is a depiction from Supplementary Video 1, and the subsequent ribbon diagrams depict a side and top view emphasising the lipid orientation. **B** Zoomed in view of the PE lipid bound simultaneously between MlaC and MlaD, emphasising how the lipid tail binds into the MlaC pocket. **C** Ribbon diagram showing a zoomed in view of the lipid, and how it is positioned relative to the central helix bundle of MlaD.

Structures of MlaFEDB have been solved in two major conformations, an outward open state, in the absence of nucleotide (PDB 7cge) and a collapsed state, in the presence of ATP and a E170Q mutation, inhibiting ATP hydrolysis (PDB 7ch0). Furthermore, GPLs have been observed in various locations: 1. In the outward open state within the central MlaE cavity, density for two GPLs have been observed, with evidence suggesting a potential extended conformation of at least one of the lipids[12,13,17]. 2. Within a cleft formed between the interfacial helix of MlaE and its adjacent TM helices[12–17]. 3. Density has been associated with the binding of detergents/lipids at various locations within the pore formed by the MlaD helix ring assembly[12–17]. These observations, together with our results presented here, allow us to propose a mechanism by which MlaFEDB functions (Fig. 9).

We hypothesise that binding of MlaC occurs during the outward open state, in the absence of ATP. GPL is already present within the MlaE cavity either through prior activity or because they have diffused into the binding pocket from the membrane.

The binding process results in ring deformation of MlaD, as reported in this study. We speculate this may alter the conformation of the MlaD TM helices and could alter the conformation of MlaE, maybe leading to cavity alteration and GPL rearrangement or stimulation of ATP binding by MlaF. Binding of ATP results in tight dimerization of MlaF (e.g. *E. coli* MlaFEDB - Walker B mutation + ATP in nanodisc[12]), leading to the collapse of the MlaE cavity, in turn resulting in squeezing of GPLs within the central cavity into the membrane via the MlaE cleft. Concurrently, we speculate that tight binding of ATP leads to a conformational change in MlaD (potentially via the transient EQTaII state observed by Chi et al.[12]) and the bound MlaC resulting in partial collapse of its GPL binding pocket and the expulsion of one of the GPL

acyl tails which becomes sequestered by the MlaD pore formed by the helix assembly (as observed through MD simulation). Such coordinated movement would prevent GPLs from the central cavity moving towards MlaC due to it already being occupied by GPLs and the MlaC cavity already being partially forced closed. ATP hydrolysis and subsequent relaxation from the collapsed state would result in reopening of the MlaE cavity and the movement of lipids from MlaCD to fill the cavity. This results in release of MlaC and the cycle able to initiate again.

This mechanism is also able to account for the observed anterograde movement of GPLs in the presence of MlaC-apo. Binding of MlaC-apo to MlaFEDB in the outward open configuration leads to conformational change in MlaC and the opening of its GPL binding cavity, as already discussed. Furthermore, we know a direct route from the membrane to MlaC must be available as evidenced by anterograde transport in the absence of ATP[4,9]. Presumably, this is via the MlaE cleft into the central cavity and subsequently up into MlaD, allowing GPLs to move from the membrane and into MlaC, driven by the high affinity MlaC has for GPLs. Indeed, this fits with our previous observations that MlaCD alone are sufficient to result in MlaC-GPL loading. This model is entirely consistent with the observation of Low et al. that showed that in the presence of ATP and repeated ATP hydrolysis no GPL loading of MlaC-apo was observed[4]. In this scenario, any GPL loading of MlaC-apo, would immediately lead to binding pocket closure and GPLs being forced back towards the MlaE-cleft and back into the membrane. Interestingly, Low et al. also observed that the binding of vanadate and ATP allowed MlaC-apo to take up lipids. We speculate that in this scenario, representative of the post-hydrolysis step (ADP-bound state), MlaFEDB can relax back into its open outward conformation, indeed

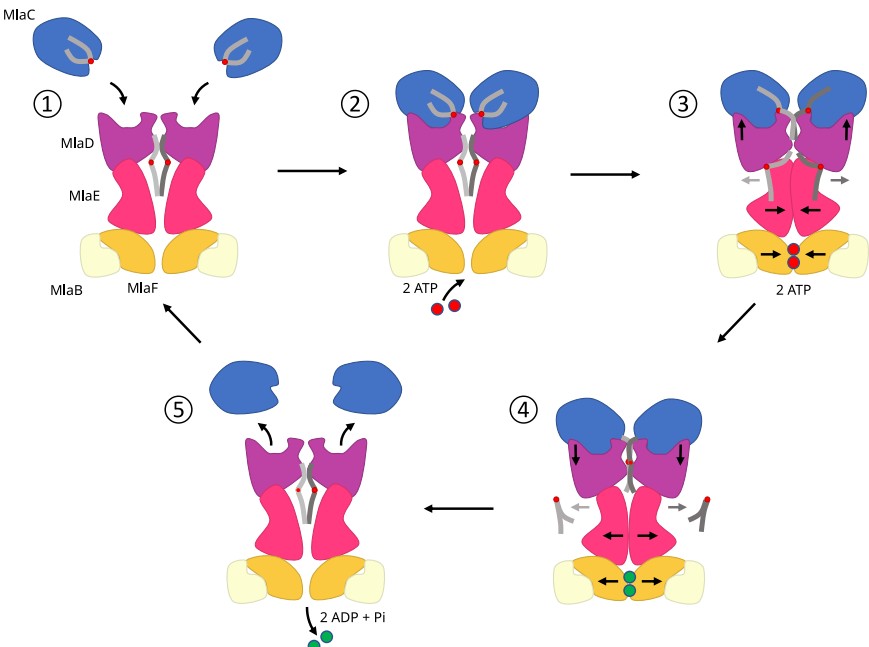

**Fig. 9 | Possible mechanism for MlaFEDB function.** (1) In the absence of ATP, MlaE (pink) is in the outward-open state forming a channel with MlaD (purple) and able to accommodate GPLs (grey) in its binding pocket. (2) MlaC-GPL (blue) is able to bind. (3) In the presence of ATP (red), the binding pocket collapses, lipids are forced out of the pocket into the membrane. Concurrently a conformational change in MlaD partially closes the MlaC binding pocket pushing a single acyl chain into the MlaD pore. (4) Following ATP hydrolysis MlaE moves back to its outward-open configuration resulting in the binding pocket opening. This conformational change drives the movement of lipids out of MlaC fully into the binding pocket. (5) MlaC-apo, ADP (green) and Pi leave allowing the cycle to start again.

the structure of this complex in the presence of ADP + vanadate has been observed (PDB 7CH8). In this conformation, GPLs from the membrane would again be able to access MlaC-apo and allow uptake by MlaC-apo.

Although we provide evidence for the mechanism by which Mla-FEDB is able to transport GPLs, additional structural information of MlaC-MlaFEDB complexes with MlaC-apo and MlaC-GPL is required to fully elucidate how ATP is utilised to drive this process in a retrograde direction. Considering lipid asymmetry is critical for barrier function, understanding retrograde-driven transport by MlaFEDB permits for the rational development of strategies to inhibit this process in the future.

## Methods

### Plasmid construction

DNA corresponding to *E. coli* MlaC was previously cloned into a custom IPTG inducible plasmid (pBE1203) (AmpR) (see ref. 18), with the leading signal sequence (residues 1–21) replaced with an N-terminal hexa-histidine tag and TEV cleavage site. A construct of the periplasmic domain of *E. coli* MlaD (residues 32–183), herein termed MlaD[32-183] with an N-terminal hexa-histidine tag was synthesised (Genscript) and cloned into the IPTG inducible pET26b (KanR) vector between the XhoI and NdeI restriction sites.

### Protein production and purification

*E. coli* BL21(DE3) cells were transformed with the relevant plasmid using heat shock. Cells were then plated on lysogeny broth (LB) agar (Melford) with resistance selection and grown at 37 °C overnight.

A single colony was picked from the overnight plate and grown at 37 °C in LB (Melford) to an $OD_{600} = 0.6$, before induction with 1 mM isopropyl β-D-1-thiogalactopyranoside and overnight expression at 18 °C. Cells were harvested by centrifugation at $6000 \times g$ for 10 minutes and resuspended in a buffer of 50 mM Tris pH 8.0, 500 mM NaCl, 10 mM imidazole supplemented with cOmplete™ EDTA-free protease inhibitors (Roche). Cells were lysed at 17,000 psi in a C3 emulsiflex cell

disruptor (Avestin) before centrifugation at $75,000 \times g$ for 45 min. The lysate was then filtered through a 0.4 μm syringe filter (Millipore) before being passed through a 5 mL HisTrap™ Ni-NTA column (Cytiva Lifesciences). The column was washed with 5 column volumes of 50 mM Tris pH 8.0, 500 mM NaCl, 10 mM imidazole (MlaC) or 50 mM Tris pH 8.0, 500 mM NaCl, 50 mM imidazole (MlaD[32-183]) before the remaining protein was eluted with 50 mM Tris pH 8.0, 500 mM NaCl, 500 mM imidazole. Protein containing fractions were pooled and gel filtered through a Superdex 75 (MlaC) or a Superdex 200 (MlaD[32-183]) size exclusion column (Cytiva Lifesciences) in a buffer of 20 mM Tris pH 8.0, 150 mM NaCl. MlaC was incubated with TEV protease overnight at a 100:1 ratio to cleave the poly-His tag before being flowed through a HisTrap™ Ni-NTA column to remove the TEV protease and uncleaved protein.

### Apo protein formation

Native protein (MlaC or MlaD[32-183]) was bound to a 5 mL HisTrap™ Ni-NTA column (Cytiva Lifesciences) and washed with 5 column volumes of 50 mM Tris pH 8.0, 500 mM NaCl, 10 mM imidazole, 50 mM n-octyl β-D-glucopyranoside (β-OG) followed by a 1 h incubation. This was repeated three times before a final wash recirculating overnight. The column was then washed with 10 column volumes of 50 mM Tris pH 8.0, 500 mM NaCl, 10 mM imidazole to remove detergent before the protein was eluted using 50 mM Tris pH 8.0, 500 mM NaCl, 500 mM imidazole before further purification by size exclusion chromatography using a Superdex 75 (MlaC) or a Superdex 200 (MlaD[32-183]) size exclusion column (Cytiva Lifesciences) in to 20 mM Tris pH 8.0, 150 mM NaCl. Confirmation of GPL removal was assessed by thin layer chromatography.

### Phospholipid loading of Mla components

Small unilamellar vesicle liposomes were prepared from chloroform solubilised cardiolipin (CL) (1′,3′-bis[1,2-dioleoyl-sn-glycero-3-phospho]-glycerol) by evaporation under a stream of nitrogen to deposit multilamellar lipid stacks. The dried lipid was resuspended to 0.45 mg/

mL in 20 mM Tris pH 8.0, 150 mM NaCl by bath sonication for 1 h. Loading of apo-MlaD[32-183] was conducted by overnight incubation of the protein with prepared liposomes at a 5-fold molar excess of GPL. MlaD[32-183] was then separated from the remaining liposomes by binding to a 5 mL HisTrap™ Ni-NTA column, washing with 5 column volumes of 50 mM Tris pH 8, 150 mM NaCl and eluted using 50 mM Tris pH 8.0, 500 mM NaCl, 500 mM imidazole followed by buffer exchange via passage through a Superdex 200 column into 20 mM Tris pH 8.0, 150 mM NaCl. Loading of apo-MlaC was conducted by 1:1 molar ratio incubation with GPL-loaded MlaD[32-183] for 1 h at 4 °C and subsequent separation by size exclusion chromatography (Superdex 200).

### Preparation of a stabilised MlaCD complex for cryo-EM

To maximise complex formation 100 µM CL loaded MlaD[32-183] was incubated with a 5x molar excess of MlaC-apo at 4 °C for 1 h. Excess MlaC as well as larger aggregates were then removed by purification on an Superdex 200 size exclusion column (Cytiva Lifescences) in 20 mM HEPES pH 7.4, 150 mM NaCl. The sample was then incubated with 0.1% (v/v) glutaraldehyde for 5 min before the cross-linking was quenched by the addition of 1 M Tris pH 8 to a final concentration of 100 mM. This cross-linked sample was then further purified on a Superdex 200 column into a buffer of 20 mM Tris pH 8.0, 150 mM NaCl.

### P1 transduction

A volume of 5 mL of LB was inoculated with an *E. coli* donor strain and grown overnight at 37 °C. 50 µL of this culture was added to 5 mL of LB supplemented with 0.2% glucose and 10 mM CaCl₂ and grown for 30 min. 100 µL of P1 phage stock at approximately $10^9$ pfu/mL was added to the culture. The culture was incubated at 37 °C for 3 h before the addition of 200 µL of chloroform. Debris was removed by centrifugation at $6000 \times g$ for 10 min and the P1 lysate was recovered. 5 mL of LB broth was inoculated with the recipient *E. coli* K12 BW25113 strain and grown overnight at 37 °C. 1.5 mL of the culture was pelleted at $6000 \times g$ and resuspended in 0.75 mL of P1 salt solution (10 mM CaCl₂/5 mM MgSO₄). 100 µL of resuspended cells were then mixed with 100 µL of P1 lysate and incubated at 37 °C for 30 min. After incubation 1 mL of LB and 200 µL of 1 M sodium citrate was added and the culture was incubated for 1 h at 37 °C. The cells were pelleted at $6000 \times g$ for 5 min before being resuspended in 100 µL of LB and plated on LB plates with Kanamycin and 5 mM sodium citrate. Plates were grown overnight and colonies were selected and re-streaked onto clean plates. Transduction was confirmed by PCR (method adapted from ref. 26).

### Complementation assay

*ΔmlaD* and *ΔmlaC* mutants were already available in the Keio library[27,28]. Gene deletions were transferred into a fresh *E. coli* K12 BW25113 strain using P1 transduction. Bacteria were grown on LB agar plates (supplemented with 5 g/L NaCl and 15 g/L of Agar) or LB medium (supplemented with 5 g/L NaCl) and incubated at 37 °C. When required, the medium was supplemented with 30 µg/mL Kanamycin or 100 µg/mL Ampicillin. Square petri dishes (120 mm × 120 mm, Merck) filled with LB agar or LB agar supplemented with 0.5% SDS, 0.5 mM EDTA and ± 2 mM TCEP. Overnight cultures were adjusted to an OD₆₀₀ of 1, and serially diluted (1 in 10) to $10^{-7}$. 2.5 µL of each dilution was pipetted onto the plate and left to dry at room temperature for 3 h prior to growth overnight at 37 °C. Complementation plates were imaged using a GelDoc system (Bio-Rad).

### Western blots for assessing the leaky expression of strains used in the complementation studies

To test leaky protein expression in the strains used for complementation studies, 5 mL cultures of *ΔmlaC* mutants containing each complementation plasmid were grown to OD₆₀₀ = 1. BW25113 and *ΔmlaC* were grown as controls. Cells were pelleted at $4000 \times g$ for

15 min and resuspended in 200 µL of SDS loading buffer (2x laemlli sample buffer, Merck). Protein expression was assessed by SDS-PAGE and western blot using an anti-MlaC primary rabbit polyclonal antibody (1:500) (Pacific Immunology, kindly supplied by Shu-sin Chng) with anti-rabbit IgG-HRP secondary (1:5000) (Goat pAb to Rb IgG (HRP) Abcam, Lot:GR3307521-1). Detection was performed using an ECL detection kit (Cytiva).

### Fluorescence assay for lipid exchange

Donor MlaA proteoliposome were created by combining *E. coli* polar lipids (EPL), NBD-tail PE and Rhodamine-PE at a ratio of 92:6:2 (wt/wt/wt) at 1 mg/mL. 0.4 mg of lipid mixture was dried down and resuspended at 0.4 mg/mL in 20 mM Tris pH 7.8, 150 mM NaCl, 3 mM ATP and 5 mM MgCl₂ followed by bath sonication for 30 min. MlaA was added directly to the sonicated lipid mixture and incubated for 30 min on ice to allow for proteoliposome formation.

Acceptor MlaFEDB proteoliposomes were formed by drying down 0.4 mg of EPL and resuspending in 20 mM Tris pH 7.8, 150 mM NaCl, 3 mM ATP and 5 mM MgCl₂ at 0.1 mg/mL then bath sonicated for 30 min. MlaFEDB was added to the sonicated EPL at 0.02 mg/mL and incubated on ice for 30 min.

For the lipid transport assay, 10 µL of MlaA donor and 10 µL MlaFEDB acceptor proteoliposomes were combined in 115 µL (20 mM Tris pH 7.8, 150 mM NaCl, 3 mM ATP and 5 mM MgCl₂) in each well of a Griener 96 flat bottom transparent microwell plate in triplicate. For lipid transport analysis 15 µL of MlaC at 0.4 mg/mL was added to the donor and acceptor proteoliposome mixture to initiate lipid transport. The increase in NBD-PE fluorescence was measured at 535 nm with a 460 nm excitation wavelength for 90 min at 37 °C.

### Cryo-EM data acquisition and data processing

Glutaraldehyde stabilised MlaCD was adjusted to 0.8 mg/mL in a buffer of 20 mM Tris pH 8.0, 150 mM NaCl. Quantifoil 300 mesh gold R2/2 holey carbon grids were glow discharged for 120 s at 40 mA. 3 µL of the sample was applied to the grid. Blotting and sample vitrification was conducted with the assistance of a Vitrobot System (ThermoFisher) then plunge frozen in a liquid ethane cryogen. Micrographs of the glutaraldehyde stabilised MlaCD[32-183] complex were recorded on a 300 kV Titan Krios microscope with a Gatan K3 Summit detector in super-resolution mode (Midlands Regional Cryo-EM facility). A total of 8741 movies at a pixel size of 0.75 Å were recorded with a total dose of 40 e⁻/Å² per 50 frames. Data processing was performed in cryoSPARC v3.3.1 (see Supplementary Fig. 2 for details). Motion correction was performed using patch motion correction (multi) and the CTF parameters were determined using patch CTF (multi). Motion corrected micrographs were manually curated, and bad micrographs excluded from further processing.

Particles were automatically picked from a subset of 500 micrographs using blob picker with a particle diameter between 120 and 150 Å using an elliptical blob and extracted with a box size of 400 pixels. These particles were used to generate representative 2D class averages and subsequently used as templates for automated particle picking using template picker on the whole data set. A total of 519,770 particles were picked and extracted using a box size of 350 pixels. The particles were passed through two rounds of 2D classification resulting in 286,001 particles from the best 2D classes.

Multi-class ab initio reconstruction was used to classify the particles in 3D using 5 classes followed by heterogeneous refinement of the initial maps. Two unambiguous configurations of the MlaCD[32-183] complex were observed from the initial and heterogeneously refined maps, with one class containing one molecule of MlaC bound to a hexameric MlaD[32-183] complex, and another class with two molecules of MlaC bound (Supplementary Fig. 2). The maps were subsequently refined using non-uniform refinement with C1 symmetry for the complex with one MlaC and C2 symmetry for the complex with two

MlaC subunits producing maps of 5.16 Å and 4.38 Å, respectively. The C1 map was further refined using local refinement with a tight mask producing a final volume at 4.35 Å. Local resolution maps were generated using local resolution estimation.

## Model building

The atomic models of the periplasmic domain of *E. coli* MlaD[32-183] (5UW2) and MlaC (5UWA) were fit into the cryo-EM maps sequentially using ChimeraX[29,30]. Residues 120–125 of MlaD comprising the partially disordered loop region between β sheets 6 and 7 were manually built in each of the six MlaD subunits using Coot[31]. The models were subjected to iterative rounds of real space refinement using PHENIX[32] and Coot. The coordinates have been deposited to the PDB under the accession numbers 8OJ4 (1:6 stoichiometry) and 8OJG (2:6 stoichiometry), and the maps have been deposited to the EMDB with the accession numbers EMD-16904 (1:6 stoichiometry) and EMD-16913 (2:6 stoichiometry).

## Coarse-grained MD simulations

For membrane-embedded simulations of the two MlaCD[32-183] complexes (1:6 & 2:6), the transmembrane regions of MlaD were added from residues 1–36 using AlphaFold[21] and aligned to our cryo-EM structure using PyMOL[33]. For simulations of the MlaFEDB-MlaC complex, MlaFEB of PDB entry 7CGE were aligned and added to both MlaCD[1-183] complexes.

All MD simulations were performed using GROMACS 2021.4[34]. The membrane-bound structures were first inserted into a preformed *E. coli* membrane using the MemProtMD pipeline[35]. The structures were oriented in a bilayer using Memembed 1.15[36] and converted to coarse-grained (CG) representations using the Martini v3.0.0 forcefield[37]. Systems contained MlaD or MlaFEDB in a single elastic network, with MlaC in a separate elastic network. Both used an upper and lower elastic bond cut-off of 1.0 nm and 0.5 nm, respectively. A preformed membrane was built with INSANE 3.0[38] with a symmetrical 80:20 PE:PG ratio, using the Memembed orientation. Non-membrane lipids were added to systems using GROMACS tools and systems were solvated with water and 0.15 M NaCl.

All systems were subjected to an energy minimisation step with the steepest descent algorithm. The systems underwent a 20 ns equilibration simulation over 0.01 ps time steps, using V-rescale group temperature coupling and semi-isotropic Berendsen pressure coupling, with respective constants of 1 ps and 12 ps. The reference temperature was 310 K and pressure 1.0 bar. The subsequent 5 µs production simulation was run over 0.02 ps steps, using the same temperature coupling as for equilibration, but with C-rescale semi-isotropic pressure coupling, set at 1.0 bar with a 12 ps constant. Both simulations used the LINCS algorithm to constrain bond lengths[39], electrostatics were modelled using reaction field and van der Waals interactions using cut-off, both with a cut-off of 1.1 nm. CG simulations were converted to atomistic (AT) representations for static visualisation using CG2AT[40,41].

## Reporting summary

Further information on research design is available in the Nature Portfolio Reporting Summary linked to this article.

## Data availability

The 1:6 structure of MlaCD generated in this study has been deposited in the PDB database under accession code 8OJ4 as well as in the EMDB database under accession code EMD-16904. The 2:6 structure of MlaCD generated in this study has been deposited in the PDB database under accession code 8OJG as well as in the EMDB database under accession code EMD-16913. The initial and final configuration of all molecular dynamics trajectories generated in this study have been deposited in the Zenodo database as 'Coarse-Grained Molecular

Dynamics simulations of Mla' [https://doi.org/10.5281/zenodo.11492165]. The Cryo-EM particle stack generated in this study has been deposited in the Electron Microscopy Public Image Archive and can be accessed through the EMDB entry of the relevant structure. The structure of MlaD[32-183] generated by Ekiert et al.[18] and used in this study is available in the PDB database under accession code 5UW2. The structure of MlaC generated by Ekiert et al.[18] and used in this study is available in the PDB database under accession code 5UWA. Source data are provided with this paper.

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

## Acknowledgements

We would like to thank the funding bodies who support us. P.W., H.J., R.H., G.R., C.B.W. and B.F.C. were jointly funded by the BBSRC and the University of Birmingham (through the Midlands Integrative Biosciences Training Partnership - Grant No. BB/M01116X/1). This work was also supported by BBSRC Research Grants No. BB/S017283/1 (T.J.K.) and BB/R019061/2 (J.R.C.B.).

## Author contributions

P.W., H.J., P.J.S., J.R.C.B. and T.J.K. co-wrote the manuscript. P.W., H.J., D.J.H., S.B., R.H., G.R., P.S., J.C., C.B.W., A.C., B.F.C., J.A.B., G.W.H., P.J.S., J.R.C.B. and T.J.K. contributed to the experimental data and data processing.

## Competing interests

The authors declare no competing interests.
