## [Peer Review File · Nature Communications]

The structure of the MlaC-MlaD complex reveals the molecular basis of periplasmic phospholipid transportREVIEWER COMMENTS

Reviewer #1 (Remarks to the Author):

This manuscript from Wotherspoon et al. investigates the lipid transport mechanism by the Mla system in *E. coli*, which likely facilitates retrograde GPL transport across the cell envelope. Little was known until very recently about how MlaC (a periplasmic lipid transfer protein) interacts with MlaD in the inner membrane to deliver substrates to the inner membrane, MlaFEDB complex. Wotherspoon et al. have been able to stabilize transiently interacting MlaC and MlaD in a complex to determine two cryo-EM structures, revealing the potential of two different complex MlaD:MlaC stoichiometries- 6:1 and 6:2. The authors used mutational analysis, genetic complementation, and FRET transport experiments to assess the functional importance of potential interaction regions based on the structure. In addition, MD simulations were used to investigate the binding/movement of lipids to the MlaCD complex.

Overall, this study is a clear step forward towards understanding lipid transport by the Mla system. A recent publication has reported similar models for the MlaCD complex based on a combination of low resolution cryo-EM data and AlphaFold predictions (PMID: 37100290). The experimental structure presented here is of modest resolution, and quite low in parts of MlaC, but we think it is still an advance or complementary to the recent publication. There is strong merit to presenting (arguably) the first experimental structures of MlaC bound to MlaD. However, some of the structural analyses lead to rather speculative conclusions not clearly supported by the data (such as whether conformational changes in MlaD are induced by MlaC binding, as opposed to many other possibilities, none of which were tested or discussed). The authors take a number of approaches to validate and probe their structures, but unfortunately, there are several shortcomings, often making it difficult to draw a clear conclusion. Thus, while there are some advances being made here, our enthusiasm is tempered by a significant number of weaknesses. Below are our suggestions to help improve the manuscript in revision:

MAJOR COMMENTS:

In 325-338: This expansion of the MlaD ring/conformational change upon MlaC binding is potentially interesting. However, there are a few potential caveats: 1) the sample used here was crosslinked, and crosslinking can stabilize alternative/non-ground state conformations of the protein; 2) MlaD was loaded with CL in the sample preparation, which might also change the conformation of MlaD; and 3) in addition to not being bound to MlaC, the 5uw2 structure also does not form the two stacked MCE rings (dodecamer) observed in the authors' sample, and ring stacking might also alter the ring conformation. To really figure out if this is due to crosslinking, CL binding, MlaD ring stacking, or MlaC binding, the authors would need to determine structures of dodecameric MlaD, with and without crosslinker (for example). We appreciate that that is probably beyond the scope of this work, but then this section needs to be adjusted to reflect what is supported by the data. While we think it is OK to mention the conformational change and the possibility that it is induced by MlaC binding, the authors should also mention the equally valid alternative possibilities to avoid misleading the reader; and the section title should be changed.

In 398-405: This key data is not shown. The data indicating that these mutants can be expressed and behave like WT should be shown.

In 460-461: An important control is missing here, showing that each single mutant (Q149C or L151C) does not impact function. If it is indeed the disulfide that is blocking function, each single mutant should function normally. There is also no data presented showing that this mutant folds normally. Based upon the data as presented, including the TCEP experiment, we think the conclusion here should be that the mutations to Cys (NOT the disulfide bond) are causing the loss of MlaD function; but that is not the conclusion made.

Transport assay: There are some questions about what is actually going on in the newly developed transport assay and how well it serves as a proxy of the underlying biology. First, the assay allows

spontaneous exchange rather than transport, in the direction opposite the stated direction of transport in cells (assay is anterograde, while stated direction for Mla in cells is retrograde). It also uses isolated MlaD periplasmic domain, instead of the functional complex, MlaFEDB. Other groups have reported transport assays using MlaFEDB in the presence of ATP, etc that seem to recapitulate retrograde transport in vitro (e.g., Tang, et al 2021; Low, et al 2021). Second, it uses a fluorescently labeled lipid (dansyl-PE) to track what is going on. Adding a fluorophore to such a small molecule has major caveats, but of course, is not uncommon in the field. However, at least one paper used radiolabeled lipids instead, which is the gold standard (Low, et al 2021). Third, the assay is FRET-based and is presumed to be due to FRET between tryptophans in MlaC and the dansyl-PE. To my knowledge, this is a completely new assay for Mla, and how this works was not rigorously tested by mutating the Trp residues to determine if a single Trp or multiple contribute to the observed signal, etc. Fourth, the authors got a strange result with the Q80E mutant, which they state they don't really understand (nor do I). Fifth, adding TCEP seemed to greatly enhance the rate of transfer/signal in the WT protein, but it seems like neither MlaD nor MlaC have Cys residues, so why this happens is not clear to me. In light of all these, we question what is really being read out here, and whether this assay is providing reliable data. We think more work is needed to validate the assay, or perhaps it should just be replaced with one of the other established assays (or the data removed?).

Fig. 6: (D) The controls and the mutants being tested are not in the same image, so it is unclear if they are from the same plate, and therefore not properly controlled. As this is a very simple experiment, we think the authors should repeat with all of the samples on a single plate.

A table with the data collection and refinement stats is missing.

MINOR COMMENTS:

In 46: Maintenance of OUTER MEMBRANE Lipid Asymmetry (Mla), according to Malinverni, et al 2009

In 66: The authors should also cite the recent pre-print/publication from MacRae and Puvanendran, et al. (PMID: 37100290), and discuss how their structures are similar or different from the models proposed there.

Unless we missed it, it isn't clearly stated in the abstract/intro/results which species the authors are studying. This is stated in the methods, but would also be good to mention somewhere early on in the main text

In 273-279: I'm not sure we entirely follow the argument here. We assume that the input MlaC is mostly loaded with CL, so why is it surprising/interesting that the unbound fraction has CL? Fig. 1D doesn't show the input MlaC pool vs fraction 1, so it is hard to know if anything changes during incubation with MlaD, but the lipids extracted from fraction 1 appear to be a mixture, is that what is meant by exchange (In 277)?

In 295: This says more than 9000, but methods say less than 9000

In 303-305: The authors observed two MlaC molecules binding symmetrically to the MlaD ring (i.e., MlaD chains B and E in their structure). Did they also observe any states with two MlaC molecules bound asymmetrically (e.g., MlaD chains B and D)? Even if these are at low resolution, it could be interesting to discuss if there are other possible binding modes, or if other modes can be ruled out based on clashes that would be expected to occur in light of their structure, etc. For example, could 6 MlaC molecules bind simultaneously? Can 3?

In 318, 328: We feel like the reported RMSDs are a bit high. For the MlaCD interface, this may reflect modest flexibility at the interface, but when comparing MlaD monomers, and 1.3-1.5 Å RMSD is more than we would expect for nearly identical monomers. We would have expected RMSDs <1.0, maybe 0.5. We would suggest double checking these; or perhaps there are differences worth discussing?

In 427: Mutations in which protein? MlaD? Not clearly stated

In 470-471: The degree of rescue here is pretty small, suggesting that the primary defect is due to one or both of the actual Cys mutants, not the disulfide bond.

In 484-486: The authors refer to a gap between helices that we don't think was mentioned before? To what are they referring? Also, this statement is not well supported by the data (e.g., Cys mutants).

In 501-504: By free lipids, do the authors mean lipids not in the bilayer ("in solution")? The depictions of the MD simulations in figure 7 are a bit hard to see. Could the authors possibly show closer and/or alternate views to help visualize the lipids in their contexts? Videos would also be very helpful in understanding. Also, the description of the simulations were confusing, making it difficult to understand the result of the simulation. Further clarification of the simulation outcomes would be beneficial.

In 508: What earlier observations are being referred to here? We feel like the authors may have forgotten to discuss something earlier on about the conformation of the MlaD C-term helices, or we missed it. It seems like a potentially interesting finding worth making more prominent.

In 565-566: This inference is not well supported and should be removed. Given that the density for some helices near the lipid binding pocket of MlaC are just featureless tubes, it isn't surprising that no CL is observed purely for data quality reasons.

In 597-600: This is quite speculative. There is no data presented here for any conformational changes in MlaC upon MlaD binding, ATP-induced or otherwise.

Fig. 1: (A) Add MW standards to plots, or indicate estimated MW of the three peaks? Peaks are numbered, but the meaning of the numbers is not mentioned in the legend. (B) For the key observation here, peak 3 + CL, the result isn't particularly compelling, though we can "just" see the three bands.

Fig. S1: (A) We are having trouble understanding what is going on in this figure. What is B and UB above lanes? Needs explanation in legend. (B) Showing MW standards would be helpful

Fig. S2: It is very hard to see maps colored by local resolution, we cannot read the numbers on the scale, and it seems to be colored in the opposite of the normal scheme. Normally blue would be highest resolution and red would be lowest, but we think the authors colored blue as lowest resolution.

Fig. 2: Arrows between panels should be made more precise to indicate the axis around which rotation occurs and direction. On the reviewer PDF, there are lines and circles surrounding each panel; presumably a glitch in the PDF?

Fig. 5: The legend needs to better describe what is shown. For example, in panels (D) and (E), which protein is being mutated? Also, while the cryo-EM structures and the MlaD mutation data suggest that the $\beta 6$ - $\beta 7$ loop on MlaD is involved in binding/function, mutating the observed interacting residues on MlaC has minimal effect. Does this mean that this region of MlaC may not be important after all? Also, for complementation assays, the labels for dilutions are missing the "-" sign (e.g., should be 10^{-3} , not 10^3)

It is very interesting that CL stabilized the MlaCD complex. What is going on there? What is the mechanism of stabilization? Do other PLs also stabilize the complex? The authors are quick to dismiss CL as a non-physiological substrate, but what is the evidence that Mla does not transport CL? It would make sense that MlaC bound to the true substrate (CL?) would have higher affinity for MlaFEDB to favor substrate delivery to MlaFEDB, then apo MlaC would have reduced affinity to facilitate release after the substrate is transferred to MlaFEDB. We think it would be worth discussing the mechanism of stabilization and substrate specificity of Mla in more detail.

While the authors do mention it in the text, we think showing the dodecameric MlaD ring in a figure somewhere is important to illustrate exactly what the sample being imaged/reconstructed is, and any caveats associated with that (i.e., the whole membrane-facing surface of MlaD could be altered by interaction with the other MlaD ring, but this ring stacking is not clearly shown anywhere in the manuscript and could easily be missed). The ring stacking and how it was dealt with is not clearly discussed at all in the data processing section of the methods.

It would be great if the authors also deposited their raw micrographs in EMPIAR so they will be available to the broader cryoEM community.

Reviewer #2 (Remarks to the Author):

This manuscript describes new insights into the Mla system, which carries out retrograde transport of phospholipids from the bacterial outer membrane to the inner membrane. The authors determine new structures of MlaC in complex with MlaD in two stoichiometries - 1:6 and 2:6. Additional experiments and simulations on the system identify key residues involved in the complex formation and function.

This paper represents an important step forward in our understanding of this complicated system. I have a few points regarding the simulations specifically that I think could be clarified or expanded upon to further improve the manuscript.

It's stated that full-length MlaD was modeled and anchored in a membrane to more accurately represent the complex. However, it's worth pointing out that MlaFEB is missing, which would normally receive the lipid from MlaD. I don't think this invalidates the observations, but they should be interpreted in this context. For example, I assume MlaD touches the membrane in these simulations, but it wouldn't do that in the full complex. This is addressed later in the brief mention of simulations of the full complex, but maybe it should be mentioned earlier as well.

It's claimed that because four lipids entering MlaD is consistent with previous results, binding of MlaC doesn't impact how many MlaD can accommodate. Why not just run the simulations of MlaD alone to confirm?

Can the expansion of the MlaD ring upon MlaC binding be recapitulated in simulations of the full complex? This would seem interesting to try, no?

Finally, 500 ns seems unusually short for coarse-grained simulations. Is there any reason the authors chose CG over atomistic simulations given the relatively short time scales investigated?

Figure 7: I think it would be helpful to see the full system, including membrane.

Figure 8: Please label the individual components in the first panel.

We respond to each of the reviewers comments in turn

Reviewer 1

Reviewer #1 – comment 1

In 325-338: This expansion of the MlaD ring/conformational change upon MlaC binding is potentially interesting. However, there are a few potential caveats: 1) the sample used here was crosslinked, and crosslinking can stabilize alternative/non-ground state conformations of the protein; 2) MlaD was loaded with CL in the sample preparation, which might also change the conformation of MlaD; and 3) in addition to not being bound to MlaC, the 5uw2 structure also does not form the two stacked MCE rings (dodecamer) observed in the authors' sample, and ring stacking might also alter the ring conformation. To really figure out if this is due to crosslinking, CL binding, MlaD ring stacking, or MlaC binding, the authors would need to determine structures of dodecameric MlaD, with and without crosslinker (for example). We appreciate that that is probably beyond the scope of this work, but then this section needs to be adjusted to reflect what is supported by the data. While we think it is OK to mention the conformational change and the possibility that it is induced by MlaC binding, the authors should also mention the equally valid alternative possibilities to avoid misleading the reader; and the section title should be changed.

Authors' Response

We attempted structure determination of dodecameric MlaD in the presence of a crosslinker. We observed sticking together of MlaD hexamers in an inhomogenous manner, please see the figure below. This suggests that the presence of MlaC prevented interactions that led to aggregative cross-linking. Unfortunately we were unable to process this data further beyond 2D class averages to give a clearer picture of the effect of crosslinking on MlaD expansion.

Because of these outcomes. We have chosen to address the reviewers comments by modifying the text to reflect the possibility of alternative causes for the conformational change of MlaD. The section title has been changed to: **“Structural changes to MlaD³²⁻¹⁸³ observed in the MlaCD complex”**

The text now reads:

“Interestingly, the largest deviations occur distal from MlaC binding, for example between MlaD monomers 2, 3 and 4. Importantly, such a break in symmetry was not observed in the structures of MlaD within the MlaFEDB inner-membrane assembly (Chi et al. 2020, Coudray et al. 2020, Zhang et al. 2020, Mann et al. 2021, Tang et al. 2021, Zhou et al. 2021), and is therefore possibly not representative of the physiological binding event. While we attribute the expansion of the MlaD ring and reorganisation of the helices to the MlaC binding event, it is equally possible that the ring expansion is due to accommodating the CL lipid, or a result of chemical crosslinking and the formation of the non-physiological dodecameric MlaD complex.”

Reviewer #1 – Comment 2

In 398-405: This key data is not shown. The data indicating that these mutants can be expressed and behave like WT should be shown.

Author’s Response

We have addressed this by providing several supplementary figures showing all relevant purifications.

The figures have been added as Supplementary Figure 5 and Supplementary Figure 6. All subsequent supplementary figures have been incremented in their legends and in text. The following txt has been added as the legends for these supplementary figures:

Supplementary Figure 5 – A) Size exclusion chromatography traces and B) SDS-PAGE of purified MlaC mutants used in this study.

Supplementary Figure 6 – Size exclusion chromatography traces and SDS-PAGE of purified MlaFEDB mutants used in this study. A) Size exclusion chromatography traces of purified MlaFEDB mutants **B)** MlaFEDB mutants, samples were not boiled before electrophoresis and bands associated with MlaD can be seen at various stages of unfolding **C)** MlaFEDB mutants, samples were boiled at 90 °C for 5 min before electrophoresis

Reviewer #1 – Comment 3

In 460-461: An important control is missing here, showing that each single mutant (Q149C or L151C) does not impact function. If it is indeed the disulfide that is blocking function, each single mutant should function normally. There is also no data presented showing that this mutant folds normally. Based upon the data as presented, including the TCEP experiment, we think the conclusion here should be that the mutations to Cys (NOT the disulfide bond) are causing the loss of MlaD function; but that is not the conclusion made.

Author's Response

We agree this should have been included. We have addressed this by making these mutants, repeating the assays and showing purifications in the supplementary. We have included this data in Figure 6 and made modifications to the text of section titled **“Access between the α 1 helices is important for activity”** to reflect the additional interpretations of the single mutants. Our results show that both Q149C and L151C under TCEP conditions are tolerated and active *in vitro*, whilst in the absence of TCEP show no activity. These results are suggestive of disulphide bond formation occurring preventing movement of lipids between the helices and that when reduced, the cysteines do not impact transport. In the cell however, the SDS/EDTA/TCEP condition is too severe to make any firm conclusions, but interestingly the Q149C mutant is tolerated in the absence of TCEP, suggesting that disulphide bond formation is unlikely to occur in the cell.

Reviewer #1 – Comment 4

Transport assay: There are some questions about what is actually going on in the newly developed transport assay and how well it serves as a proxy of the underlying biology. First, the assay allows spontaneous exchange rather than transport, in the direction opposite the stated direction of transport in cells (assay is anterograde, while stated direction for Mla in cells is retrograde). It also uses isolated MlaD periplasmic domain, instead of the functional complex, MlaFEDB. Other groups have reported transport assays using MlaFEDB in the presence of ATP, etc that seem to recapitulate retrograde transport *in vitro* (e.g., Tang, et al 2021; Low, et al 2021). Second, it uses a fluorescently labeled lipid (dansyl-PE) to track what is going on. Adding a fluorophore to such a small molecule has major caveats, but of course, is not uncommon in the field. However, at least one paper used radiolabeled lipids instead, which is the gold standard (Low, et al 2021). Third, the assay is FRET-based and is presumed to be due to FRET between tryptophans in MlaC and the dansyl-PE. To my knowledge, this is a completely new assay for Mla, and how this works was not rigorously tested by mutating the Trp residues to determine if a single Trp or multiple contribute to the observed signal, etc. Fourth, the authors got a strange result with the Q80E mutant, which they state they don't really understand (nor do I). Fifth, adding TCEP seemed to

greatly enhance the rate of transfer/signal in the WT protein, but it seems like neither MlaD nor MlaC have Cys residues, so why this happens is not clear to me. In light of all these, we question what is really being read out here, and whether this assay is providing reliable data. We think more work is needed to validate the assay, or perhaps it should just be replaced with one of the other established assays (or the data removed?).

Authors' Response

We have repeated all functional analysis using the assay developed by Tang et al. 2021 to address the reviewers concerns. We have made modifications to the methods section and results section to reflect the change to methodology and results. Overall the results are consistent with our previous assay.

Reviewer #1 – Comment 5

Fig. 6: (D) The controls and the mutants being tested are not in the same image, so it is unclear if they are from the same plate, and therefore not properly controlled. As this is a very simple experiment, we think the authors should repeat with all of the samples on a single plate.

Authors' Response

The results were on the same plate but due to ordering of mutations they were in separate areas. We have now repeated these experiments and placed relevant mutations together for ease of viewing. The complete western blots are shown in Supplementary Figure 8.

Reviewer #1 – Comment 6

A table with the data collection and refinement stats is missing.

Authors' Response

Apologies, this was accidentally missed in the draft we first submitted. We have now included this in the Supplementary as Supplementary Table 1 and renumbered other tables accordingly.

Reviewer #1 – Comment 7

In 46: Maintenance of OUTER MEMBRANE Lipid Asymmetry (Mla), according to Malinverni, et al 2009

Authors' response

We have now corrected this.

Reviewer #1 – Comment 8

In 66: The authors should also cite the recent pre-print/publication from MacRae and Puvanendran, et al. (PMID: 37100290), and discuss how their structures are similar or different from the models proposed there.

Authors' Response

We have addressed the recent MacRae paper in the introduction. The text now reads:

“However, the molecular details of the interaction between MlaFEDB and MlaC have, until recently, remained elusive. Ercan et al. attempted to map this interaction through crosslinking experiments (Ercan et al. 2018), and a more detailed understanding of the interaction has only recently been elucidated (MacRae et al.), with the assistance of AlphaFold2 predictions. Although, due to the simulated nature of this model and the low-resolution of the actual structural data presented there is still a large degree of uncertainty regarding the nature of the MlaFEDB to MlaC interaction.”

We have also made modifications to the discussion as follows:

“Finally, we have identified the MlaD β 6- β 7 loop region as a significant site of interaction between MlaC and MlaD, with mutations within this loop, which complexes at the back of MlaC distal from the MlaC binding site, completely negating GPL transport and function of the complex *in vivo*. This same region was predicted using alphafold modelling by MacRae *et al.* (2023), they then tested the effect of mutation on binding affinity and observed mutations within this interaction site to significantly impact binding affinity between MlaC and MlaD. The position of this loop interaction, at the back of MlaC, close to the pivot of the β -sheet, alludes to a mechanism by which it may control the opening/closing of the MlaC GPL binding cavity. First consider anterograde transport observed by us and others. We have shown here and previously that MlaC-apo is unable to take up GPLs from the environment. However, binding to MlaD, between the β 6- β 7 loop and the main body could pull the GPL binding cavity open allowing GPLs to enter. This is consistent with our structures of MlaCD, which show MlaC in an open configuration bound to MlaD. In the case of retrograde transport, a conformational change in MlaD, as a result of ATP binding, could lead to either the upwards movement of β 6- β 7 whilst the main body of MlaD remains fixed, or vice versa, leading to the closing of the MlaC cavity and the expulsion of GPLs contained within.”

Reviewer #1 – Comment 10

Unless we missed it, it isn't clearly stated in the abstract/intro/results which species the authors are studying. This is stated in the methods, but would also be good to mention somewhere early on in the main text.

Authors' Response

We have now addressed this by making reference to *E. coli* in both the abstract and introduction.

The text now reads:

“Here, we report the structure of *E. coli* MlaC in complex with the MlaD hexamer in two distinct stoichiometries.”

And

“In this study, we have addressed this knowledge gap. Specifically, we report the structure of *E. coli* MlaC in complex with the MlaD hexamer, in two different stoichiometries (1:6 and 2:6, respectively), stabilised through the binding of cardiolipin”

Reviewer #1 – Comment 11

In 273-279: I'm not sure we entirely follow the argument here. We assume that the input MlaC is mostly loaded with CL, so why is it surprising/interesting that the unbound fraction has CL? Fig. 1D doesn't show the input MlaC pool vs fraction 1, so it is hard to know if anything changes during incubation with MlaD, but the lipids extracted from fraction 1 appear to be a mixture, is that what is meant by exchange (In 277)?

Authors response

It is the MlaD that is loaded with cardiolipin not MlaC and as such it is interesting that although MlaC gets bound stably to MlaD, some MlaC does manage to get separated and is loaded with CL. However, for clarity we have adjusted the text as follows:

“Although stabilisation was observed, some MlaC was able to be separated from MlaD³²⁻¹⁸³-CL and showed the presence of cardiolipin bound (Figure 1D). However, compared to PG and PE exchanged between natively prepared MlaD³²⁻¹⁸³ and MlaC-apo (Ercan et al. 2018, Hughes et al. 2019), the observed exchange was minimal. Furthermore, it remains unclear whether this was a true binding event, with CL bound within the central cavity, or just loosely associated. Overall, the observed stabilisation of the MlaC-MlaD³²⁻¹⁸³ complex (thereafter termed MlaCD³²⁻¹⁸³) in the presence of CL and the decreased capacity for CL exchange suggests that within the cell CL is unlikely to be a natural substrate for the Mla pathway as this stabilisation likely impedes transport rates.”

Reviewer #1 – Comment 12

In 303-305: The authors observed two MlaC molecules binding symmetrically to the MlaD ring (i.e., MlaD chains B and E in their structure). Did they also observe any states with two MlaC molecules bound asymmetrically (e.g., MlaD chains B and D)? Even if these are at low resolution, it could be interesting to discuss if there are other possible binding modes, or if other modes can be ruled out based on clashes that would be expected to occur in light of their structure, etc. For example, could 6 MlaC molecules bind simultaneously? Can 3?

Authors Response

We found no evidence of any other binding modes. We saw either a single MlaC bound or two MlaC's bound on opposite sides of the MlaD hexamer. No others were evident. To make this clearer in the text we have adjusted it to read:

“Further 2D and 3D classification of the data did not reveal any evidence for particles with additional copies of MlaC bound to the MlaD³²⁻¹⁸³ hexamer, or for any MlaCD(2:6) structures with the two MlaC monomers bound in any orientation other than directly opposite each other. However, it is yet to be resolved whether 1 or 2 MlaC monomers per MlaD hexamer is the native biological assembly. Data presented by MacRae et al. (2023) would suggest that orientations where MlaC binds to residues that are not directly opposite do exist, however, this may be an artefact of their methodology of stabilising the interaction by increasing avidity with a trimerised MlaC construct to promote multivalent binding. Our structure, supplemented by the prior crosslinking study carried out by Ercan et al. (2018) suggests that there is involvement of the MlaD monomers adjacent to the primary binding monomer as shown in Figure 4, which, along with direct steric hindrance between MlaC molecules, would likely interfere with binding of MlaC at these loci. MacRae et al. also suggested that binding would be unlikely to occur at the n+1 or n-1 positions, however, their data suggests that binding at the n+2 position is possible. While, we agree that there would not be steric hindrance between the binding MlaC molecules in this arrangement, the lack of evidence for such a binding orientation in our data combined with several other considerations, such as the breakdown of the 2-fold symmetry of the MlaFEDB complex, the necessity for dual involvement of the n+1 MlaD monomer in two binding events, and the nature of the structures presented by MacRae et al., which appears to be more of a transient docking of MlaC to the $\beta 6$ - $\beta 7$ loop rather than a binding event permitting functional lipid exchange lead us to conclude that the binding orientations we have observed are the only orientations that permit lipid exchange *in vivo*.”

Reviewer #1 – Comment 13

In 318, 328: We feel like the reported RMSDs are a bit high. For the MlaCD interface, this may reflect modest flexibility at the interface, but when comparing MlaD

monomers, and 1.3-1.5 Å RMSD is more than we would expect for nearly identical monomers. We would have expected RMSDs <1.0, maybe 0.5. We would suggest double checking these; or perhaps there are differences worth discussing?

Authors Response

We had initially wanted to do a somewhat surface level comparison here, by taking the MlaD1 protomer and determining its RMSD comparison over the other protomers in the model, to show an overall minimal change to the general structure of each protomer. However, in retrospect this approach may have been inappropriate and while the quoted RMSDs of 1.3 - 1.5 are technically correct for the comparison between MlaD1 and the non-adjacent protomers this may have been an overly simplified analysis and we appreciate the opportunity to correct our error in judgement here. We have adjusted the text to read as follows:

“Within the limitations of the map’s resolution, pairwise comparison between MlaD protomers within the MlaD hexamer suggest that all six copies of the protein adopt similar conformations in the core fold (residues 40 - 141). MlaCD(1:6) has a C α RMSD of between 0.7 - 0.8 Å in the core fold across all pairwise protomer alignments. Likewise, MlaCD(2:6) has a C α RMSD of between 0.7 - 0.9 Å in the core fold across all pairwise protomer alignments, except for those positioned directly opposite in the hexamer, which have RMSDs as low as 0.3 Å in the core fold and 0.4 Å for the entire C α trace. The MlaD_{1,4} and the MlaD_{3,6} pairs have significantly lower RMSDs for the full C α trace, likely owing to the interaction with MlaC restricting positional variability in the central helix and β 6- β 7 loop. In contrast, pairwise comparison of the adjacent MlaD_{1,6} pairs in both the MlaCD(1:6) and MlaCD(2:6) structures and the MlaD_{3,4} pair in the MlaCD(2:6) structure have an RMSD of 1.8 Å for the full C α trace owing to differential interactions of the central helices with MlaC resulting in them moving apart, creating somewhat of a gap between the helices of these monomers. RMSDs for the pairwise comparison of the total C α trace for the remaining MlaD pairs falls in the range of 0.9 - 1.2 Å for the MlaCD(1:6) structure and 1.3 - 1.5 Å for the MlaCD(2:6) structure.

Similarly, comparison with the previously published crystal structure of MlaD³²⁻¹⁸³ in isolation (5uw2) shows that each MlaD monomer of MlaD³²⁻¹⁸³ undergoes little conformational change upon binding to MlaC, with each monomer having an overall C α RMSD of 1.3-1.4 Å between the apo- and MlaC-bound states, with the major structural deviations being in the central helix and β 6- β 7 loop. However, although conformational changes between the core fold of monomers were minimal, comparison of the MlaD hexameric architecture between the apo- and MlaC-bound states revealed a striking rearrangement of the MlaD³²⁻¹⁸³ hexamer, with a clear expansion of the ring observed (Figure 3A).”

Reviewer #1 – Comment 14

In 427: Mutations in which protein? MlaD? Not clearly stated.

Authors Response

We have adjusted the text so it more clearly highlights the mutations are in MlaD. It now reads:

“Analysis of MlaD F118K, E119K, D120K and E122K...”

Reviewer #1 – Comment 15

In 470-471: The degree of rescue here is pretty small, suggesting that the primary defect is due to one or both of the actual Cys mutants, not the disulfide bond.

Authors Response

We have already addressed this as part of our response to Reviewer #1 – Comment 3.

Reviewer #1 – Comment 16

In 484-486: The authors refer to a gap between helices that we don't think was mentioned before? To what are they referring? Also, this statement is not well supported by the data (e.g., Cys mutants).

Authors Response

We have addressed this as part of our response to Reviewer #1 - Comment 13, a small gap between the helices of MlaD1,6 and MlaD3,4 is observed.

Reviewer #1 – Comment 17

In 501-504: By free lipids, do the authors mean lipids not in the bilayer ("in solution")? The depictions of the MD simulations in figure 7 are a bit hard to see. Could the authors possibly show closer and/or alternate views to help visualize the lipids in their contexts? Videos would also be very helpful in understanding. Also, the description of the simulations were confusing, making it difficult to understand the result of the simulation. Further clarification of the simulation outcomes would be beneficial.

Authors Response

Free lipids are referring to lipids that are not initially in the bilayer at the beginning of the simulation. We have clarified what we mean by Free lipids within the text this now reads

“free lipids (Mix of PE & PG randomly placed within the experimental frame)”

We have also updated figure 7 and separated it into two new figures (figure 7 & 8) with alternative views as well as alternative depictions to hopefully better show how the lipids bind in the pocket. We have also included several videos to the same effect. In essence, the simulations suggest that lipid transport into and out of the proteins seems to be handled one acyl chain at a time. We have clarified this as the take home message of the simulations as follows:

In summary, all observed cases of lipid transport occurred through the sequential exchange of a single acyl tail.

Reviewer #1 – Comment 18

In 508: What earlier observations are being referred to here? We feel like the authors may have forgotten to discuss something earlier on about the conformation of the MlaD C-term helices, or we missed it. It seems like a potentially interesting finding worth making more prominent.

Authors Response

We realise the language used could have been mis-interpreted, as such we have adjusted the text to now read:

“Furthermore, during repeat simulations we also observed the occasional simultaneous binding of a single PE GPL between both MlaC and MlaD, each bound via a single acyl tail (Figure 7D & 7E). Binding was between the $\alpha 1$ helices of MlaD₁ and MlaD₆, displacing them from their canonical position lining the central MlaD helix bundle”

Highlighting we repeated simulations and have moved the reference to Figures 7D & 7E earlier so it more clearly references the figure.

Reviewer #1 – Comment 19

In 565-566: This inference is not well supported and should be removed. Given that the density for some helices near the lipid binding pocket of MlaC are just featureless tubes, it isn't surprising that no CL is observed purely for data quality reasons.

Authors Response

We have removed reference to the lack of CL. The text now reads:

“This is consistent with our structures of MlaCD, which show MlaC in an open configuration bound to MlaD”

Reviewer #1 – Comment 20

In 597-600: This is quite speculative. There is no data presented here for any conformational changes in MlaC upon MlaD binding, ATP-induced or otherwise.

Authors Response

We agree it is speculative, no-one has yet to determine the complete structure of MlaFEDB with MlaC bound. We state we are speculating clearly in the text and as such do not believe it needs changing. The text reads:

“Concurrently, we **speculate** that tight binding of ATP leads to a conformational change in MlaD (potentially via the transient EQTall state observed by (Chi et al. 2020)) and the bound MlaC resulting in partial collapse of its GPL binding pocket and the expulsion of one of the GPL acyl tails which becomes sequestered by the MlaD pore formed by the helix assembly (as observed through MD simulation)”

Reviewer #1 – Comment 21

Fig. 1: (A) Add MW standards to plots, or indicate estimated MW of the three peaks? Peaks are numbered, but the meaning of the numbers is not mentioned in the legend. (B) For the key observation here, peak 3 + CL, the result isn't particularly compelling, though we can “just” see the three bands.

Authors Response

We have updated Figure 1 to include the MW standards. We have also updated the main text and figure legend as follows:

Main text

“Purification by size exclusion chromatography yielded separate species suggesting low binding affinity (Figure 1A - peaks 1 & 2 left panel & Figure 1B). However, by first removing all lipid (as we have performed previously (Hughes et al. 2019)) then incubating MlaD³²⁻¹⁸³ with cardiolipin, we found that following co-incubation with MlaC-apo, the two proteins formed a stable complex (Figure 1A - peak 2 right panel & Figure 1B). Complexation was further validated by analytical ultracentrifugation, which showed a clear difference in the sedimentation coefficient between the individual components (MlaC-apo, MlaD³²⁻¹⁸³-apo & MlaD³²⁻¹⁸³-CL) and the MlaC-apo:MlaD³²⁻¹⁸³-CL complex (Figure 1C).

Figure legend

“Figure 1 – Cardiolipin stabilizes the MlaCD complex A) Size exclusion chromatography (Superdex 200) analysis of MlaC and MlaD and their copurification in the absence (CL-) and presence (CL+) of cardiolipin. Peak numbers are indicated above the peaks. Peak 1 refers to MlaC elution, Peak 2 MlaD elution and Peak 3 aggregate. In the presence of cardiolipin, the ratio of Peak 1 to Peak 2 is altered showing increased protein in Peak 2. B) SDS-PAGE of peak fractions from A) showing the presence of MlaC co-purifying with MlaD in Peak 2 in the presence of cardiolipin only. C) AUC sedimentation velocity analysis of the effect of cardiolipin on MlaCD

complex formation. D) Thin layer chromatography showing the presence of cardiolipin within the MlaC fraction isolated from A).”

Reviewer #1 – Comment 22

Fig. S1: (A) We are having trouble understanding what is going on in this figure. What is B and UB above lanes? Needs explanation in legend. (B) Showing MW standards would be helpful.

Authors Response

We have updated the figure legend to explain that B refers to the sample that has been boiled and UB refers to a sample that has not been boiled prior to SDS-PAGE. This is usually to show full denaturation of the MlaD hexamer which when unboiled is sufficiently stable to run as a hexamer to ~100 kDa. In this case it shows the sample has been sufficiently crosslinked. We have also added MW standards to the SEC trace.

The legend now reads:

A) SDS-PAGE of the effect of glutaraldehyde exposure on MlaCD complex stabilisation under boiled (B) and unboiled (UB) conditions.

Reviewer #1 – Comment 23

Fig. S2: It is very hard to see maps colored by local resolution, we cannot read the numbers on the scale, and it seems to be colored in the opposite of the normal scheme. Normally blue would be highest resolution and red would be lowest, but we think the authors colored blue as lowest resolution.

Authors Response

We have modified the figure to make the maps coloured by local resolution larger and the colour has been modified to be in line with convention. The figure caption has been modified to read:

“Supplementary Figure 2 - Processing pipeline of the MlaCD dataset in cryoSPARC v3.3.1.”

Reviewer #1 – Comment 23

Fig. 2: Arrows between panels should be made more precise to indicate the axis around which rotation occurs and direction. On the reviewer PDF, there are lines and circles surrounding each panel; presumably a glitch in the PDF?

Authors Response

We have modified the figure to more precisely show the axes around which rotation occurs and the direction of rotation.

Reviewer #1 – Comment 24

Fig. 5: The legend needs to better describe what is shown. For example, in panels (D) and (E), which protein is being mutated? Also, while the cryo-EM structures and the MlaD mutation data suggest that the $\beta 6$ - $\beta 7$ loop on MlaD is involved in binding/function, mutating the observed interacting residues on MlaC has minimal effect. Does this mean that this region of MlaC may not be important after all? Also, for complementation assays, the labels for dilutions are missing the “-” sign (e.g., should be 10^{-3} , not 10^3)

Authors Response

We have updated this accordingly. The legend now reads as follows:

“Figure 5 – The $\beta 6$ - $\beta 7$ loop is essential for function. A) Screen for SDS/EDTA sensitivity of cells carrying pET22b encoding the WT or mutated copies of MlaC or MlaD in the parent or $\Delta mlaC / \Delta mlaD$ strain background. WT – BW25113 parent strain. Cells were normalised to an OD_{600} of 1 and 10-fold serially diluted before being spotted on LB agar containing the indicated condition. Western blot showing levels of MlaC within the cell. **B)** The positioning of the $\beta 6$ - $\beta 7$ loop interacting with MlaC is shown, highlighting the residues mutated in A). **C- & D)** FRET based GPL transport assays. Fluorescence increase corresponds to a reduction in NBD-PE FRET quenching by Rhodamine-PE as lipids are transferred from the MlaA proteoliposome to the MlaFEDB proteoliposome. Excitation wavelength 460 nm, emission wavelength 535 nm. The traces in C) correspond to the MlaD mutants investigated in this study alongside the positive WT control and the $\Delta MlaFEDB$ negative. The traces in D) correspond to the MlaC mutants with mutations to residues proximal to the $\beta 6$ - $\beta 7$ loop of MlaD during interaction, alongside the positive WT control and the $\Delta MlaFEDB$ negative.”

We have also updated our discussion of mutations within MlaC based on our updates to the assay. This has already been discussed in response to Reviewer #1 - comment 3.

Reviewer #1 – Comment 25

It is very interesting that CL stabilized the MlaCD complex. What is going on there? What is the mechanism of stabilization? Do other PLs also stabilize the complex? The

authors are quick to dismiss CL as a non-physiological substrate, but what is the evidence that Mla does not transport CL? It would make sense that MlaC bound to the true substrate (CL?) would have higher affinity for MlaFEDB to favor substrate delivery to MlaFEDB, then apo MlaC would have reduced affinity to facilitate release after the substrate is transferred to MlaFEDB. We think it would be worth discussing the mechanism of stabilization and substrate specificity of Mla in more detail.

Authors Response

We do not have any clear evidence for why CL is stabilizing the complex as the CL lipid is unresolved in our structures. Our best guess would be that 2 acyl chains of the CL lipid bind in MlaC and 2 acyl chains bind in MlaD, preventing dissociation of MlaC and stalling the interaction in the state where the acyl tail of PG/PE lipids would be exchanged. We chose not to include this assessment due to lack of any structural evidence. With regards to CL as a substrate, our observations here are not sufficient to claim that CL is not a substrate. But as there is clear evidence in the literature that CL is not found bound to natively purified MlaC (Fig S3 of Ercan 2018 is one example), we were simply suggesting that our observations here support the lack of any pre-existing evidence for natively purified CL bound MlaC. To clarify this we have changed the text as follows:

“Overall, the observed stabilisation of the MlaC-MlaD³²⁻¹⁸³ complex (thereafter termed MlaCD³²⁻¹⁸³) in the presence of CL and the decreased capacity for CL exchange, alongside an observed lack of CL bound MlaC in literature (Ercan et al. 2018), suggests that within the cell CL is unlikely to be a natural substrate for the Mla pathway as this stabilisation likely impedes transport rates.”

Reviewer #1 – Comment 26

While the authors do mention it in the text, we think showing the dodecameric MlaD ring in a figure somewhere is important to illustrate exactly what the sample being imaged/reconstructed is, and any caveats associated with that (i.e., the whole membrane-facing surface of MlaD could be altered by interaction with the other MlaD ring, but this ring stacking is not clearly shown anywhere in the manuscript and could easily be missed). The ring stacking and how it was dealt with is not clearly discussed at all in the data processing section of the methods.

Authors Response

We did not do a reconstruction of the dodecameric ring. This is because the interface is not the same across particles, and therefore it is not a homogeneous dodecamer. It would not be possible to reconstruct the dodecameric structure with sufficient resolution to determine specific interactions between the two membrane-facing surfaces.

These 2D classes taken from Supplementary Figure 2 illustrate this:

These, as well as the *ab initio* classes in Supplementary Figure 2 should illustrate the true particle being reconstructed.

Reviewer #1 – Comment 27

It would be great if the authors also deposited their raw micrographs in EMPIAR so they will be available to the broader cryoEM community.

Authors Response

The particle stack has been deposited with accession number **47485412**. We have also now included a data availability section at the end of the paper.

Reviewer #2 – Comment 1

It's stated that full-length MlaD was modelled and anchored in a membrane to more accurately represent the complex. However, it's worth pointing out that MlaFEB is missing, which would normally receive the lipid from MlaD. I don't think this invalidates the observations, but they should be interpreted in this context. For example, I assume MlaD touches the membrane in these simulations, but it wouldn't do that in the full complex. This is addressed later in the brief mention of simulations of the full complex, but maybe it should be mentioned earlier as well.

Authors Response

We have addressed this by stating that the observations are intended to provide insight into the dynamics of the interaction and lipid exchange between MlaD and MlaC in the absence of the MlaFEB complex as the opening premise of the section. The text now reads as follows:

To provide additional insight into the dynamics of binding and lipid exchange between MlaC and MlaD in the absence, and then subsequently the presence of the MlaFEB complex, we conducted a series of MD simulations investigating the movement of lipid within the stabilized MlaCD complex we had generated. While this does not provide direct insight into the mechanism of function of the MlaFEDB complex it does assist in understanding observations made when investigating the function of lipid exchange between MlaC and soluble MlaD.

Reviewer #2 – Comment 2

It's claimed that because four lipids entering MlaD is consistent with previous results, binding of MlaC doesn't impact how many MlaD can accommodate. Why not just run the simulations of MlaD alone to confirm?

Authors Response

We did conduct simulations with a GPL bilayer and MlaD alone, but did not see a difference in the number of bound lipids. We chose to compare our observations to the results of Thong et al. because they provided validation in literature for our results and our results provide insight into the nature of the binding that Thong et al. had observed. As such, we deemed inclusion of the MlaD simulations redundant, however, upon re-examination it is clear we have made a claim here that we did not provide sufficient support for. As such, we have included results from these simulations as videos and have modified the text as follows:

The binding of a maximum of 4 lipids is consistent with the observations of Thong et al. (2016) as well as our own observations with MlaD alone (Supplementary Video 3) and suggests the binding of MlaC does not impact the number of lipids MlaD can accommodate (Thong et al. 2016). While these observations do not inform the default binding state of MlaD as part of the MlaFEDB complex, they do give insight into the dynamics of the bound state observed by Thong et al. and inform us on the maximal binding capacity of MlaD.

Reviewer #2 – Comment 3

Can the expansion of the MlaD ring upon MlaC binding be recapitulated in simulations of the full complex? This would seem interesting to try, no?

Authors Response

We did not observe reorganisation of the central helix in response to MlaC binding, however, significant reorganisation of the central helix does occur in Supplementary Video # in response to the binding of free lipids to MlaD, we have addressed this in text as follows:

MD simulations of the interactions between MlaC and MlaD in the presence of lipids seem to suggest that the binding of MlaC does not cause reorganisation of the MlaD central helix, however, the interaction between MlaD and lipids alone does cause significant reorganisation (Supplementary Video 3). Although, as we were unable to resolve any lipids in our structural data we cannot confirm if the reorganisation we observe in the MlaCD (1:6) and (2:6) structures is due to the passage of lipids between the helices.

Reviewer #2 – Comment 4

Finally, 500 ns seems unusually short for coarse-grained simulations. Is there any reason the authors chose CG over atomistic simulations given the relatively short time scales investigated?

Authors Response

All simulations have been extended to 5 μ s (25 repeats), however, no notable differences were observed beyond the original 500 ns endpoint. We therefore retain the original figures that depict examples of lipid binding. The text has been updated to reflect the new simulation length and reads 5 μ s at all points that referenced the old 500 ns simulation length. The text of Figure 7 has been updated to read:

B) Example snapshot of MlaCD (1:6) at 500 ns of the 5 μ s simulation, showing 4x PE (2x membrane lipids; green and beige).

D) Example snapshot of of MlaCD (1:6) at 500 ns of the 5 μ s simulation, showing PE binding simultaneously to MlaC and MlaD(1:6), each via a single tail, between the α 1 helices of MlaD.

Reviewer #2 – Comment 5

Figure 7: I think it would be helpful to see the full system, including membrane.

Authors Response

We have addressed this issue as part of the response to Reviewer #1 – Comment 17 and have modified the figure to include the membrane as well as including videos of the simulations as supplementary material.

Reviewer #2 – Comment 6

Figure 8: Please label the individual components in the first panel.

Authors Response

We have updated the figure with this detail

REVIEWERS' COMMENTS

Reviewer #1 (Remarks to the Author):

I think the authors have adequately addressed my concerns.

Reviewer #2 (Remarks to the Author):

The authors have sufficiently addressed all of my comments.